# Amending the algorithm of aerosol-radiation interaction in WRF-Chem (v4.4)

Jiawang Feng[1], Chun Zhao[1,2,3], Qiuyan Du[1], Zining Yang[1], Chen Jin[1]

[1] Deep Space Exploration Laboratory/School of Earth and Space Sciences, University of Science and Technology of China, Hefei, China
[2] Laoshan Laboratory, Qingdao, China
[3] CAS Center for Excellence in Comparative Planetology, University of Science and Technology of China, Hefei, China

*Correspondence to*: Chun Zhao (chunzhao@ustc.edu.cn)

**Abstract.** WRF-Chem is widely used to assess regional aerosol radiative feedback. However, in current version, aerosol optical properties are only calculated in four shortwave bands, and only two of them are used to "interpolate" optical properties towards 14 shortwave bands used in the Rapid Radiative Transfer Model (RRTMG) scheme. In this study, we use a "Resolved" algorithm to estimate aerosol radiative feedback in WRF-Chem, in which aerosol optical properties are calculated in all 14 shortwave bands. The impacts of changing this calculation algorithm are then evaluated. The simulation results of aerosol optical properties are quite different using the new "Resolved" algorithm, especially for dust aerosols. The alteration of aerosol optical properties result in considerably different aerosol radiative effects: the dust radiative forcing in the atmosphere simulated by the "Resovled" algorithm is about two times larger than the original "Interpolated" algorithm; The dust radiative forcing at top of the atmosphere (TOA) simulated by the "Interpolated" algorithm is negative in all Sahara region, while the "Resolved" algorithm simulates positive forcing at TOA and can exceed $10\,\mathrm{W\,m^{-2}}$ in the Sahara desert, which is more consistent with previous studies. The modification also leads to changes in meteorological fields due to alterations in radiative feedback effects of aerosols. The near-surface temperature is changed due to the difference in radiation budget at the bottom of the atmosphere (BOT) and the heating effects by aerosols at the surface. Furthermore, the amendment of algorithm partially corrects the wind field and temperature simulation bias compared to the reanalysis data. The difference in planet boundary layer height can reach up to ~100 m in China and ~200 m in Sahara, further resulting in a greater surface haze considerably. The results show that correcting the estimation algorithm of aerosol radiative effects is necessary in WRF-Chem model.

## 1 Introduction

Aerosol-radiation interaction and its impacts on meteorological processes and aerosol cycle have been proven to be important (e.g., Ackerman, 1977; Dickerson et al., 1997; Jacobson, 1998; Zhao et al., 2010, 2011, 2012, 2013, 2014; Myhre et al., 2013; Bender et al., 2020; Bellouin et al., 2020; Huang and Ding, 2021). As the two-way interaction between aerosol

and meteorological fields are complex, a fully coupled "online" meteorology-chemistry model is a necessary tool to account for these feedbacks in simulating aerosol concentrations and meteorological fields. WRF-Chem (the Weather Research and Forecasting model coupled with Chemistry) is one of the most widely used atmospheric models that consider aerosol-radiation interactions for investigating regional aerosol lifecycle and radiative impacts (e.g., Zhao et al., 2010, 2011, 2013,

2014; Jiang et al., 2012; Ding et al., 2013; Wu et al., 2013; Gao et al., 2014; Chen et al., 2014; Zhong et al., 2016; Liu, et al., 2016; Huang et al., 2016; Petäjä et al., 2016; Du et al., 2020, 2023; Zhang et al, 2020; Wang et al., 2022; Chen et al., 2022; Sharma et al., 2023; Wei et al., 2023). WRF-Chem is capable of performing regional-scale simulations with high spatial resolution. This allows for a detailed representation of aerosol and radiation processes at regional scale. By incorporating the interactions between aerosols and radiation, WRF-Chem can provide insights into the impacts of aerosols on regional

weather patterns, climate, air quality, and the energy balance. Therefore, it is crucial to simulate appropriately aerosol optical properties and then aerosol-radiation interaction in WRF-Chem for the modelling community.

In WRF-Chem, aerosol optical properties (i.e., aerosol optical depth (AOD), single scattering albedo (SSA), and asymmetry factor) for shortwave are first computed for four wavelengths of 300, 400, 600, and 999 nm following the method as described in previous studies (Fast et al., 2006; Barnard et al., 2010; Zhao et al., 2013) (more details in Sect. 2.2).

Afterwards, these aerosol optical properties are used in radiative transfer schemes such as RRTMG (Mlawer et al., 1997; Iacono et al., 2000; Zhao et al., 2011). In shortwave bands, RRTMG calculates radiative fluxes and heating rates in fourteen bands of the shortwave. However, due to that the aerosol optical properties for shortwave are only calculated for four spectral bands as mentioned above, the RRTMG scheme interpolates the values at these four wavelengths to fourteen wavelengths to be used. For AOD, the scheme obtains the values for all fourteen shortwave bands using the Ångström

exponent (Ångström, 1929) based on AOD at 400 nm and 600 nm. For SSA and asymmetry factor, simple linear interpolation is applied (Barnard et al., 2010; Zhao et al., 2011). These interpolation methods of aerosol optical properties of two bands into fourteen bands may lead to significant errors in estimating aerosol radiative forcing and subsequently simulating aerosol radiative feedback on meteorological fields.

Therefore, in this study, we amend the aforementioned approach by calculating directly aerosol optical properties for all

fourteen RRTMG shortwave bands, and examine the difference between the new and original algorithms on simulating multi-band aerosol optical properties, radiative forcing, and aerosol impacts on meteorological fields. The paper is organized as follows. Section 2.1 briefly introduces the WRF-Chem, and Sect. 2.2 describes the algorithms in simulating aerosol-radiation interaction in WRF-Chem. The difference in simulating aerosols optical properties and radiative impacts with the two algorithms are investigated in Sect. 3. The conclusion and summary are in Sect. 4.

## 2 Methodology

### 2.1 WRF-Chem

The WRF-Chem model is a version of WRF model (Skamarock et al., 2021) that simulates trace gases and particulates simultaneously with the meteorological fields (Grell et al., 2005). The Model for Simulating Aerosol Interactions and Chemistry (MOSAIC) aerosol model (Zaveri et al., 2008) and Carbon Bond Mechanism (CBMZ) photochemical mechanism (Zaveri and Peters, 1999) implemented by Fast et al. (2006) into WRF-Chem, which includes complex treatments of aerosol radiative properties and photolysis rates, are used in this study. Since this study only focuses on the amendment of current algorithm of aerosol optical properties and its radiative feedback in WRF-Chem, more details about physics and chemistry schemes in WRF-Chem are not described here and can be found in previous studies (Zhao et al., 2011, 2013).

### 2.2 Amendment of algorithm of aerosol-radiation interaction in WRF-Chem

In current (v4.4) and previous versions of WRF-Chem, aerosol optical properties such as extinction, SSA, and asymmetry factor for scattering are computed as a function of wavelength and three-dimensional position. Currently, the methodology described by Ghan et al. (2001) is applied to compute the extinction efficiency $Q_e$ and the scattering efficiency $Q_s$ in WRF-Chem. In the model, the full Mie calculation is performed only once to obtain a table of seven sets of Chebyshev expansion coefficients, and later the full Mie calculations are skipped and $Q_e$ and $Q_s$ are calculated using bilinear interpolation over the Chebyshev coefficients stored in the table. The detailed method of the computation of aerosol optical properties in the model is similar to the description in Fast et al. (2006), Barnard et al. (2010), and Zhao et al. (2013). The Optical Properties of Aerosols and Clouds (OPAC) dataset (Hess et al., 1998) is used for the shortwave (SW) and longwave (LW) refractive indices of dust aerosols. Radiative feedback of aerosols is coupled with RRTMG (Mlawer et al., 1997; Iacono et al., 2000) for both longwave (LW) and shortwave (SW) radiation as Zhao et al. (2011). AOD and direct radiative forcing of aerosols are diagnosed following the methodology by Zhao et al. (2013). In this methodology, the calculation of aerosol optical properties and radiative transfer scheme is performed multiple times with the mass of one or more aerosol species (i.e., the mass of an individual or a group of aerosol species) and also its associated water aerosol mass removed from the calculation each time. After this diagnostic procedure, the optical properties (e.g., AOD) and direct radiative forcing for aerosols can be estimated by subtracting the optical properties and direct radiative forcing from the diagnostic iterations from those estimated following the standard procedure for all the aerosol species. It can be described as:

$$AOD_{[species\ i]} = AOD_{[all\ species]} - AOD_{[without\ species\ i]} \tag{1}$$

$$AForcing_{[species\ i]} = Forcing_{[all\ species]} - Forcing_{[without\ species\ i]} \tag{2}$$

Currently, the aerosol optical properties for shortwave spectral (roughly 200 nm ~ 4000 nm) are only calculated for four

shortwave wave bands centered at 300, 400, 600, and 999 nm following the method above. When coupled with RRTMG shortwave radiative transfer scheme (RRTMG-SW) in WRF-Chem, the aerosol optical properties (i.e., AOD, SSA, and asymmetry factor) need to be interpolated from the values at the four wave bands to those values at the fourteen wave bands (from 232 nm to 3462 nm) used in the RRTMG shortwave scheme. The interpolation of AOD is based on the Ångström exponent that is derived from the values at 400 nm and 600 nm:

$$\alpha = -\ln\left(\frac{\tau_{600}}{\tau_{400}}\right)/\ln\left(\frac{600}{400}\right) \tag{3}$$

Where $\alpha$ is the Ångström exponent, $\tau_{400}$ and $\tau_{600}$ are the AOD over 400 nm and 600 nm bands.

Given the Ångström exponent, AOD over wave band centered at $\lambda_i$ is calculated as:

$$\tau_i = \tau_{600}\left(\frac{\lambda_i}{600}\right)^{-\alpha}, (i = 1, 2, \dots, 14) \tag{4}$$

The SSA and asymmetry factor for other wave bands are linearly interpolated from the values at the four wave bands. This current algorithm of calculating optical properties and coupling with the shortwave radiation transfer scheme is referred to as "Interpolated Algorithm" in the rest of this paper.

In this study, the "Resolved Algorithm" that the aerosol optical properties over the fourteen short wave bands are calculated and coupled with RRTMG-SW directly is implemented. While our study focused on the sectional representation of aerosols, our amendment of algorithm is valid for bulk, sectional and modal representations in WRF-Chem, theoretically. However, we have not tested the differences between "Interpolated" and "Resolved" methods for bulk and modal aerosol representations in this study. The difference between "Interpolated" algorithm and "Resolved" algorithm is defined as the bias due the interpolation of aerosol optical properties for radiation transfer scheme. The biases of the "Interpolated" algorithm and its impacts are investigated.

## 2.3 Numerical experiments

In this study, four sets of experiments are conducted over two domains as shown in Fig. 1. One covers China (8°N-55°N, 59°E-146°E) that represents the region with complex aerosol sources including large anthropogenic aerosol mass loading and also natural dust over the Northwest. The other covers Sahara (3.5°S-42°N,24°W-44°E) that represents the region with the largest natural dust aerosol mass loading of the world. In addition, Fig. 1a and Fig. 1b also delineate regions dominated by anthropogenic aerosols and dust aerosols, respectively, using dashed-line boxes (referred to as anthro-dominant and dust-dominant regions). Over both domains, two sets of experiments, one with the "Interpolated" algorithm and the other with the "Resovled" algorithm, are conducted. The simulations are performed at 50 km × 50 km horizontal resolution with 120 × 100 grid cells and 40 vertical layers up to 100 hPa. The model time step is set at 150 seconds, and the aerosol optical properties are updated every 30 minutes in the model. The experiments are conducted for January and July of 2015 representing boreal winter and summer, which starts from 25 December 2014 to 31 January 2015 and from 25 June 2015 to 31 July 2015, respectively. Only the results during January and July are used in the analysis to minimize the impact from the chemical

initial conditions. In this study, all analysis results, unless otherwise stated, represent monthly averages for January and July 2015. The meteorological initial conditions are derived from the European Centre for Medium-Range Weather Forecast (ERA5) reanalysis dataset at approximately 25 km horizontal resolution and 6-hour temporal interval (Dee et al., 2011). The chemical lateral boundary conditions are from the quasi-global simulation with 360×145 grid cells (180°W~180°E,67.5°S~77.5°N) at the 1°×1° horizontal resolution (Zhao et al., 2013; Hu et al., 2016). Besides the aforementioned sets of experiments, four sets of sensitive experiments with aerosol radiative feedback disabled are also conducted to examine the aerosol radiative feedback effects on meteorological fields. We also conducted a detailed analysis of the computational costs associated with both the "Resolved" and "Interpolated" methods. For the "Resolved" method, the calculation of aerosol optical properties takes 11,458.6 seconds, which accounts for 9.3% of the total simulation runtime of 122,717 seconds. In contrast, the "Interpolated" method requires 5,223.05 seconds for the same calculations, which accounts for 4.9% of its total runtime of 107,615 seconds. Therefore, the "Resolved" method takes approximately 2.19 times computational cost for aerosol optical property calculations compared to the "Interpolated" method. This difference in calculation time translates to an additional 14% in total simulation runtime when using the "Resolved" method. It's worth noting that this increase in computational cost is less than the 3.5 times one might expect from increasing the number of shortwave bands from 4 to 14. This is because the aerosol optical properties process includes both shortwave and longwave calculations. The original WRF-Chem has already used "Resolved" method to calculate the longwave part, therefore, we only modified the shortwave calculations from 4 bands to 14 bands.

Anthropogenic emission for the domain covering China is from the Multi-resolution Emission Inventory for China (MEIC) at 0.1° × 0.1° horizontal resolution for 2015 (Li M et al., 2017a, b), while the one for the domain covering Sahara is obtained from the Hemispheric Transport of Air Pollution version-2 (HTAPv2) at 0.1° × 0.1° horizontal resolution for 2010 (Janssens-Maenhout et al., 2015). The dust emission flux is calculated with the GOCART dust emission scheme (Ginoux et al., 2001), and the size distribution of emitted dust particles follows a theoretical expression based on the physics of scale-invariant fragmentation of brittle materials derived by Kok (2011). The detailed aerosols size distribution from approximately 0.04 μm to 10 μm are listed in Table 1. More details about the dust emission scheme in WRF-Chem can be found in Zhao et al. (2010, 2013). Biomass burning emissions are obtained from the Fire Inventory from NCAR (FINN) with hourly temporal resolution and 1 km horizontal resolution (Wiedinmyer et al., 2011). Sea-salt emission follows Zhao et al. (2013), which includes correction of particles with radius less than 0.2 μm (Gong, 2003) and dependence of sea-salt emission on sea surface temperature (Jaeglé et al., 2011). The detailed parameterization schemes of physical and chemical processes of the WRF-Chem model used in the study are summarized in Table 2.

## 2.4 Dataset

To evaluate the modelling results, multi-spectral AOD measurements are required, which is critical for this study comparing the "Interpolated" and "Resolved" methods across wavelengths. Therefore, we retrieved total AOD from the AERONET

network (Holben et al., 1998). Although satellite AOD products such as from the Moderate Resolution Imaging Spectroradiometer (MODIS) have greater spatial coverage, the number of wavelengths is limited. Comparison at only one or very few wavelengths would not effectively demonstrate the distinctions between the two algorithms examined in this work. In this study, the monthly mean AOD from the AERONET Version 3 Direct Sun Algorithm, Level 2.0 dataset is used. A subset of stations dominated by dust is selected. The selected stations need to meet the following conditions: (1) to reduce

the impact of oceanic aerosols, only the sites that are located on land are used; (2) to compared with the simulation results, the sites must contain data for January and July in 2015; and (3) the monthly average Ångström Exponent (AE, 500 nm - 870 nm) at each site should be less than 0.8 because the lower the AE is, the larger the dust fraction (Dubovik et al., 2002).

## 3 Results

### 3.1 Impacts on aerosol optical properties

AOD is one of the key optical properties of aerosol. The simulated AOD over anthro-dominant regions in China and AOD over dust-dominant regions in Sahara at different wavelengths with the two algorithms are shown in Fig. 2. The results presented are spatial averages over the regions delineated by the dashed boxes in Fig. 1, as well as temporal averages for the months of January and July 2015. The blue line represents the simulated AOD for fourteen shortwave bands used by the RRTMG radiation scheme with the "Interpolated" algorithm based on the calculated values for 400 nm and 600 nm

wavelengths; The red line represents the simulated AOD for fourteen shortwave bands directly calculated with the "Resolved" algorithm. As seen in Fig. 2a, both the "Resolved" and "Interpolated" algorithms produce similar exponential decaying trends in AOD for regions dominated by anthropogenic aerosols. The AOD values calculated with the "Resolved" algorithm are slightly higher than those obtained with the "Interpolated" algorithm. These trends indicate that Ångström's theory is applicable to a certain extent in these areas. However, results in Fig. 2b illustrate a significant impact of algorithm

modification on the simulation of AOD in regions dominated by dust aerosols. Both the "Resolved" and "Interpolated" algorithms calculated an upward trend at 400 nm and 600 nm wavelengths. However, since the "Interpolated" algorithm only includes information from these two wavelengths, it results in an exponential increase across all bands. Conversely, the "Resolved" algorithm reveals a fluctuating downward trend for dust aerosols across all wavelengths. This discrepancy leads to a substantial difference at longer wavelengths, with the maximum divergence exceeding 50%. To assess whether the

"Resolved" algorithm is more accurate, we compared our simulated AOD results with the AERONET observations in dust-dominant regions of the Sahara, as shown in Fig. 3. The AERONET stations are selected follow the algorithm described in Sect. 2.4. The geographical locations of these stations are marked in Fig. 1b. AERONET observations demonstrate a gradual decline in dust AOD with increasing wavelength, with minimal variation between 400 nm and 600 nm. Although both the "Resolved" and "Interpolated" algorithms incorrectly predict an increasing trend in AOD between 400 nm and 600 nm, the

"Resolved" method's more detailed spectral optical calculations offer the ability of correcting this discrepancy at other wavelengths. Conversely, the "Interpolated" algorithm's erroneous trend at these wavelengths, when applied in the Ångström

exponent theory, propagates and amplifies discrepancies at longer wavelengths. These findings underscore the need for cautious application of the Ångström exponent theory, particularly when dealing with aerosols such as dust, whose optical properties exhibit minimal variation at the selected wavelengths (400 nm and 600 nm in WRF-Chem). In such cases, small uncertainties can lead to a reversal in the AOD-wavelength relationship, potentially resulting in significant simulating errors in spectral extrapolation. Additionally, the simulated total AOD was also compared with the AERONET results in anthro-dominant regions of China (see Supplementary Fig. S1). The results demonstrate that both the "Interpolated" and "Resolved" algorithm simulations exhibit significant decreasing trends in total AOD as those seen in the AERONET data. Therefore, the discrepancies introduced by the application of the Ångström exponent theory has much smaller effects on simulating results, which further indicates that the amendments to the algorithm have less impact on simulated AOD resulting from anthropogenic aerosols compared to dust aerosols.

Another crucial aerosol optical property is SSA, which is illustrated in Fig. 4. The amendment of algorithm results in smaller anthropogenic and dust SSA over a considerable range of wavelength, which could result in an overall larger absorption effect. Smaller than 600 nm waveband, the SSA simulated by "Resolved" algorithm generally follows a linear function as suggested by "Interpolated" algorithm in both regions. However, the value of SSA no longer increases with the increase in wavelength when it reaches around 600 nm. Moreover, beyond 2000 nm in wavelength, the "Resolved" algorithm starts to decrease, whereas the "Interpolated" algorithm does not simulate this pattern. Figure 5 shows the aerosol absorption optical depth (AAOD) as a function of wavelength. The AAOD is calculated by subtracting the scattering radiation from the extinction radiation (AOD) by aerosols, which can be described as $AAOD = AOD * (1 - SSA)$. AAOD can represent the absorption effects caused by aerosols in the atmosphere. Although the "Resolved" algorithm simulates smaller anthropogenic AOD in China and dust AOD in Sahara, the "Resolved" algorithm simulates larger AAOD at all fourteen bands due to its simulated smaller SSA. In WRF-Chem, water content of aerosols is also considered while calculating optical properties. Water exhibits a peak in absorption at approximately 2900 nm, which significantly influences the overall aerosol optical properties at this wavelength. As illustrated in Fig. S2 of the supplementary materials, there is a pronounced peak in the imaginary part of the refractive index of water at around 2900 nm in "Resolved" method. The imaginary part of the refractive index is directly related to absorption, with higher values indicating stronger absorption. This peak in water absorption leads to the observed increase in AAOD (decrease in SSA) at this wavelength. In contrast, the "Interpolated" method, which relies solely on optical property information at 400 nm and 600 nm, fails to capture this crucial spectral feature. Consequently, the "Interpolated" method is unable to accurately represent the complex wavelength-dependent optical properties of aerosols. Besides, the lower SSA values and higher AAOD values obtained from the "Resolved" method compared to the "Interpolated" method in the Sahara region are primarily due to the spectral variation of dust optical properties, particularly at shorter wavelengths. In OPAC dataset, dust has a larger imaginary part of the refractive index at wavelengths shorter than ~600nm, indicating stronger absorption in this spectral range (also as shown in Fig. S2). However, in the latest version of WRF-Chem ("Interpolated" method), the imaginary part of the refractive index for dust is set to a constant value of 0.003 at all four shortwave bands. The "Resolved" method simulates higher AAOD by using

the wavelength-dependent refractive indices, especially at shorter wavelengths. This results in significant differences in SSA and AAOD between the two methods, particularly in dust-dominated regions like the Sahara.

The differences in AAOD indicates that the algorithm modifications lead to a larger absorption effect from aerosols, which will be illustrated more detailed in the discussions below (see Sect. 3.2 and 3.3).

## 3.2 Impacts on radiative forcing of aerosols

As discussed above, there are significant differences in the aerosol optical properties computed based on these two algorithms, which may also lead to biases in simulating aerosol radiative forcings. Radiative forcing is defined as the perturbation of radiative fluxes at TOA and BOT, as well as the perturbation of radiative heating/cooling in the atmosphere (ATM) if a specific aerosol species is removed. It should be noted that the aerosol radiative forcing calculated in this section refers to the change in radiative fluxes resulting from the removal of aerosols in a single experiment (see Sect. 2.2), excluding the perturbations to other meteorological variables caused by the removal of aerosol radiative effects as introduced by Zhao et al. (2013). In this study, the net downward radiative flux at TOA and BOT is considered positive, while upward flux is considered negative; the heating effect of radiative flux within the ATM is considered positive, while the cooling effect is considered negative. Figure 6 illustrates the spatial distribution of aerosol radiative forcing computed using the "Interpolated" and "Resolved" algorithms, as well as the differences between the two algorithms at TOA, BOT, and ATM in China. Figure 7 shows the aerosol radiative forcing in Sahara. In both China and Sahara, the "Resolved" algorithm simulates more aerosol "warming" effects in ATM, more negative radiative forcing at BOT, and smaller negative forcing at TOA compared to the "Interpolated" algorithm. These discrepancies can be explained by the stronger aerosols' absorption effects computed from the "Resolved" algorithm, as discussed in Sect. 3.1.

As shown in Fig. 6, in regions predominantly influenced by anthropogenic aerosols, the algorithm amendment primarily affects the aerosol radiative forcing in ATM. As discussed in Sect. 3.1, the "Resolved" algorithm, compared to the "Interpolated" algorithm, is capable of simulating stronger aerosol absorption effects, resulting in a stronger "heating" effect in ATM (approximately 30% enhancement). Additionally, the algorithm amendment introduces more pronounced "cooling" effects at BOT due to the aerosol radiative perturbations. The combined effects in ATM and at BOT contribute to a relatively smaller impact of algorithm amendment on aerosol radiative forcing at TOA. On the other hand, in the northwestern part of China, which includes the Gobi Desert and the Taklimakan Desert (two major dust source regions), aerosols are predominantly composed of dust. In these regions, the impacts of algorithm amendment on radiative forcing are more prominent compared to the impacts in anthro-dominant regions. In the Sahara region, the "Resolved" algorithm predicts a much greater dust "warming" effect in ATM compared to the "Interpolated" algorithm (approximately 140% higher). Furthermore, the "Interpolated" algorithm only simulates negative dust radiative forcing at TOA in dust-dominant areas. In contrast, the "Resolved" algorithm can simulate positive forcing at TOA and can exceed $10 \mathrm{~W~m^{-2}}$ in the Sahara Desert due to the radiative absorption of dust when located over highly reflective surfaces. The positive aerosol radiative forcing at TOA in the Sahara region, simulated by the "Resolved" algorithm, is notably more consistent with previous studies (e.g.,

Albani et al., 2014; Feng et al., 2022; Feng et al., 2023) compared to the "Interpolated" algorithm. Furthermore, our previous study (Feng et al., 2023) initially employed the original optical calculation algorithm in WRF-Chem (referred to as the "Interpolated" algorithm), which yielded a dust direct radiative forcing at TOA of -0.75 W m$^{-2}$. This value significantly deviated from the observationally constrained estimate of -0.20 W m$^{-2}$ proposed by Kok et al. (2017). Subsequently, in the final version of Feng et al. (2023), we implemented the "Resolved" method, which resulted in a substantially improved estimate of -0.27 W m$^{-2}$. This marked improvement in alignment with observational constraints strongly suggests that the "Resolved" method demonstrates better performance in simulating radiative forcing, particularly for dust aerosols. Apart from TOA and ATM, the algorithm amendment also results in a stronger negative radiative forcing (approximately 50% higher) of dust aerosols at BOT, which may lead to cooling effects at the Earth's surface. In summary, the modified algorithm has an effect on anthropogenic aerosol radiative forcing, albeit relatively small. However, its impact on dust aerosols is significantly pronounced, to the extent of yielding divergent outcomes at TOA. Hence, to enhance the accuracy of aerosol radiative feedback simulation in the model, the amendment of this algorithm is imperative.

Figure 8 displays the vertical profiles of the shortwave aerosol heating rates calculated by the two algorithms over anthro-dominant areas in China and dust-dominant areas in Sahara, respectively. In both regions, the shortwave heating effect induced by aerosols is strongest at the surface and decreases rapidly as the altitude approaches approximately 500 hPa. Beyond this altitude, the aerosol-induced heating effect stabilizes at a relatively constant value, maintaining a positive heating effect throughout the atmospheric column. In the areas predominantly affected by anthropogenic aerosols, the "Resolved" algorithm simulates a more pronounced aerosol heating effect, displaying a vertical trend similar to that produced by the "Interpolation" algorithm. In the dust-dominant area, however, the impact of algorithmic amendment on the simulation of aerosol heating effects is significantly greater than that for anthropogenic aerosols, with the heating rate near the surface exceeding twice the result simulated by the "Interpolated" algorithm. This discrepancy diminishes near the altitude of 500 hPa but remains approximately 80% stronger than that predicted by the "Interpolation" algorithm. This also explains the impact of algorithm amendment on the simulation of radiative forcing in ATM, as discussed in previous. The results of heating profile also illustrates that the effects of algorithm amendment vary significantly among different types of aerosols.

It should be noted that: the refractive indices of dust can vary significantly depending on mineral composition and source region. Our study uses a set of refractive indices that represent a global average, which may not capture the full range of variability observed in different regions. Changing these values could affect the magnitude of the radiative forcing differences between the "Interpolated" and "Resolved" simulations. We have conducted additional sets of experiments with smaller radiative indices for dust aerosols in "Resolved" simulations. The results show that with less absorbing dust (smaller imaginary part of refractive indices), the differences of radiative effects could become smaller (not shown here). Theoretically, one could identify a set of aerosol optical properties that would yield similar aerosol radiative forcing results for the "Resolved" and "Interpolated" algorithms. However, such a set might not accurately reflect the true physical properties of the aerosols. Despite these considerations, it is important to note that the spectral variation of optical properties

would be better captured by the "Resolved" method compared to the "Interpolated" method as discussed in Sect. 3.2. This provides a more robust foundation for accurately representing the complex interactions between aerosols and radiation
across various wavelengths. The advantages of the "Resolved" method in capturing these spectral variations persist regardless of the specific refractive indices employed in the simulations.

## 3.3 Impacts on radiative effects of aerosols

The impacts of the algorithm amendment on radiative forcing of aerosols can further influence the radiative feedback of aerosols on meteorological fields, such as temperature, wind field, and planetary boundary layer (PBL) height as discussed
below. An additional set of experiments with radiative feedback of aerosols disabled are conducted. The differences in simulation results between the two sets of experiments (one with aerosol radiative feedbacks enabled and the other with them disabled) are used assess the radiative impacts of aerosols on meteorological fields.

Figure 9 shows the aerosol radiative effects on 2-m temperature from the "Interpolated" and the "Resolved" experiments, as well as the differences between the two algorithms. From Fig. 9a and 9b, it can be observed that in regions dominated by
anthropogenic aerosols in China, aerosol radiative effects lead to surface cooling in both algorithms. In the Tibetan Plateau region, it results in surface warming. According to the results in Fig. 6c and 6f, aerosols exhibit a cooling effect on the surface due to their direct radiative effect in all regions. This difference indicates that the radiative effects of aerosols on near-surface temperature are influenced not only by radiative forcing (direct radiative effect) but also by other factors such as aerosol-cloud interactions, the aerosols heating effects and the radiative effects on the surface radiative fluxes discussed
below. Figure 10 illustrates the impacts of aerosols on the shortwave radiative fluxes at BOT. Compared with Fig. 6, these effects include not only the direct radiative effect (DRE) of aerosols but also the indirect consequences: the alterations in radiative forcing induced by aerosols can lead to effects on the atmospheric energy balance, which in turn influence other meteorological processes such as the cloud formation and thus further affecting radiation. Figures 10a and 10b exhibit similar spatial distribution characteristics compared to Fig. 9a and Fig. 9b, indicating that the radiative effects of aerosols on
2-m temperature is primarily influenced by their effects on surface radiative fluxes.  In addition, 2-m temperature is also affected by the aerosol heating near the surface. As shown in Fig. 8a, the new "Resolved" algorithm simulates a stronger heating near the surface compared to the "Interpolated" algorithm. This turns out that the average aerosol effects on 2-m temperature are slightly larger with the "Resolved" algorithm than with the "Interpolated" algorithm, although average surface radiative cooling is stronger in the "Resolved" algorithm.

In the Sahara region, aerosol effects on radiative fluxes at the surface is similar to the aerosol radiative forcing (Fig. 7c/f), showing a cooling effect at the surface throughout the domain (Fig. 9d/e). It is noteworthy that although the aerosol effects on radiative fluxes at the surface is negative in Sahara region, the effects on 2-m temperature with both algorithms could exhibit warming effect. This is due to the aerosol heating effect in the atmosphere near the surface, which also leads to stronger warming effect of aerosols at the surface from the "Resolved" algorithm in some areas. In both regions, the aerosol
effects simulated with the new "Resolved" algorithm reduce the simulation biases in 2-m temperature with the "Interpolated"

algorithm to some extent, compared to the ERA5 data, in particular over the Sahara region (Fig. S3 in the supporting material).

The aerosol effects on the winds and geopotential heights at 850 hPa simulated by both algorithms are illustrated in Fig. 11. As the results indicate, in eastern China, the aerosol radiative effects simulated with the "Interpolated" algorithm leads to a significant decrease in geopotential height, while the effects are relatively small with the "Resolved" algorithm. Interestingly, over the Sahara region, the aerosol effects on geopotential heights and wind fields at 850 hPa simulated by the "Interpolated" algorithm are small, while the effects simulated by the "Resolved" algorithm are much larger. This may be due to the significantly larger reduction and warming effects from the "Resolved" algorithm over the Sahara region. Distinct mechanisms of aerosol effects on wind fields and geopotential heights from different type aerosols over different climate regimes deserve further investigation in future. In both regions, the algorithm amendment results in significant differences in aerosol effects on wind fields, which partially reduces the biases in simulated cyclonic wind circulation compared to the ERA5 reanalysis along the southeastern coastal region of China (Fig. S4 in the supporting material).

Previous studies have also highlighted the important role of aerosol effects on the development of PBL and hence on the air quality near the surface (e.g., Liu et al., 2016; Wilcox et al., 2016; Yang et al., 2017). Aerosols could reduce the near-surface temperature and also heat the atmosphere upper as discussed above, and therefore, suppress the PBL development (Huang et al., 2018). Figure 12 illustrates the aerosol radiative effects on PBL heights and their differences between the simulations with the "Resolved" and "Interpolated" algorithms. In the regions dominated by anthropogenic aerosols in China, the heating rates in the upper or around the PBL top due to absorbing aerosols are relatively small (Fig. 8a). Therefore, the difference in aerosol effects on PBL height between the two algorithms are primarily associated with their difference in 2-m temperature (Fig. 9). However, in the Sahara region, the difference of aerosol effects on 2-m temperature, as well as the heating rates at the upper or around the PBL top, is significant between the two algorithms (Fig. 8 and Fig. 9). Overall, aerosol effects suppress PBL development in most areas of Sahara, leading to a decrease of PBL height. Consequently, the algorithm amendment significantly affects the aerosol effects on PBL development in both regions, with an average reduction of ~20 m and up to ~100 m at maximum in China and with an average reduction of ~40 m and up to the maximum of ~200 m in the Sahara.

As previously mentioned, aerosol radiative effects on the height of PBL could concurrently affect air pollutant concentrations within the PBL (Ding et al., 2016). Figure 13 illustrates the aerosol radiative effects on PM10 (particulate matter with diameters 10 μm and smaller) at surface and the differences between the simulations with the two algorithms. In the China region, Fig. 13 shows similar spatial patterns as Fig. 12, confirming that lower PBL can raise the surface PM concentration. In the Sahara region, while aerosol radiative effects generally lead to an increase in surface PM10 concentrations, there are still areas with reduction, particularly for the simulations with the "Interpolated" algorithm. The "Resolved" algorithm results in significant differences in the effects on surface PM10 in Sahara. Please note, the impacts over Sahara are more complex because aerosol radiative effects could affect both PBL heights and emissions (through near-surface wind) and hence the near-surface mass concentrations.

## 4. Summary and discussion

Aerosol-radiation interaction can have important impacts on meteorological processes and aerosol cycle. The WRF-Chem model as a fully coupled "online" meteorology-chemistry model has been widely used to investigate the impacts of aerosol-radiation interaction at regional scale. In this study, the original "Interpolated" algorithm for calculating aerosol optical properties and radiative effects in WRF-Chem is re-examined against the "Resolved" algorithm implemented in this study. Two domains are selected for investigating the difference between the two algorithms, with one covering China that represents the region with complex aerosol sources including large anthropogenic aerosol mass loading and also natural dust over the Northwest and the other covering Sahara that represents the region with the largest natural dust aerosol mass loading of the world.

The discrepancies between the two algorithms show distinct regional characteristics. In China, where anthropogenic sources dominate the aerosol composition, the differences between the "Resolved" and "Interpolated" algorithms are relatively small, which could potentially be indicative of similar patterns in other regions dominated by anthropogenic aerosols. In contrast, the Sahara Desert, which is dominated by dust aerosols, exhibits significant differences between the two algorithms: The "Resolved" results of dust AOD shows a general downward trend with increasing wavelength, rather than an upward trend calculated by Ångström exponent. The maximum difference between the two algorithms can reach about 50%. Further comparison with AERONET observations reveals that the "Resolved" algorithm's AOD simulations are in better agreement with the measured values at dust-dominant stations. This suggests that the "Resolved" approach can more accurately capture the optical properties of dust aerosols. The "Resolved" algorithm also simulates smaller SSA than the "Interpolated" algorithm. Affected by these two factors (AOD and SSA), the "Resolved" algorithm simulates larger AAOD than the "Interpolated" algorithm, resulting in larger aerosols' heating effects.

The impacts of algorithm amendment on aerosol radiative forcing are different depending on the aerosol type and region. For areas in China with high concentrations of anthropogenic aerosols, the "Resolved" algorithm enhances the aerosol radiative absorption in the atmosphere by about 30%, compared to the "Interpolated" algorithm. It also introduces larger cooling effect at the surface. The impact on the radiative forcing at the top of atmosphere is small. In the areas dominated by dust aerosol, the impacts of algorithm amendment are substantially larger. The "Resolved" algorithm predicts a ~140% higher warming effect in the atmosphere from dust aerosol compared to the "Interpolated" algorithm. Moreover, the "Resolved" algorithm can simulate positive radiative forcing at the top of atmosphere (exceed $10\,\mathrm{W\,m^{-2}}$) in dust-dominant areas, aligning better with previous studies constrained by observations, against the negative values with the "Interpolated" algorithm. The algorithm amendment also causes a roughly 50% larger negative radiative forcing at the surface from dust, leading to stronger surface cooling.

Besides the impacts on aerosol optical properties and radiative forcing, the impacts on aerosol radiative effects on meteorological fields are also investigated. Both algorithms simulate that aerosols reduce (increase) near-surface temperature in the anthro-dominant areas (Tibetan Plateau) of China. The "Resolved" algorithm leads to slightly larger increase of near-

surface temperature in China than the "Interpolated" algorithm. Over the Sahara region, both algorithms simulate dominant cooling effect on near-surface temperature over most region but with warming effect over some areas. The "Resolved" algorithm leads to stronger effects in either cooling or warming areas. The difference of aerosol effects on near-surface temperature between the two algorithms can be explained from their difference in simulating net radiative fluxes at the surface that can be resulted from both aerosol direct (radiation) and indirect (cloud) effects. The experiment with the "Resolved" algorithm simulates better the 2-m temperature compared with the ERA5 reanalysis data than the one with the "Interpolated" algorithm. Additionally, the algorithm amendment also leads to different aerosol effects on the wind fields and geopotential height over both regions with larger impact over the Sahara compared to over China. Both algorithms simulate the aerosol radiative effects to suppress the PBL development and thus reduce the PBL height. The algorithm amendment leads to a further reduction of PBL height of ~20 m in China and ~40 m in the Sahara on domain average, respectively. The enhancement of aerosol radiative effects on reducing the PBL height by the "Resolved" algorithm leads to more accumulation of surface concentrations of PM10.

Please note that the impacts of aerosol-radiation interaction on meteorological fields and chemical species are not only through the direct effects on radiation but also through indirect effects on cloud and then precipitation and winds. For example, some difference between the two algorithms in simulating surface PM10 concentration may not be fully explained by their difference in radiative fluxes but also from the contribution from other factors such as their induced changes in surface wind driven emissions and precipitation driven wet removals (Feng et al., 2023). More details about analyzing the mechanisms driving the difference between the impacts of two algorithms deserve further investigation in future. This study underscores the importance of refining the algorithm of aerosol-radiation interaction for simulating aerosol effects on weather and climate more accurately. It cautions the usage of original "Interpolated" algorithm in WRF-Chem for simulating aerosol optical properties and their impacts on meteorological fields, which has some biases particularly for the regions with large contribution from dust. It is necessary to update the model to use the new "Resolved" algorithm proposed in this study in future.

It's important to note that the primary aim of our study is to demonstrate the importance of using a spectrally resolved method for calculating aerosol optical properties, rather than to improve the overall model performance. Many factors can affect the simulation of meteorological fields and radiative processes beyond the optical properties methods we're investigating in this study. For example, while our study employs a 50 km grid resolution, which is suitable for investigating aerosol-radiation interactions, higher-resolution could enhance the simulation of aerosol emission, deposition, and transport processes, potentially leading to a more accurate representation of aerosol distributions and their radiative effects (Feng et, al., 2023; Tan et, al., 2015; Tao et, al., 2020). In addition to model resolution, the calibration factors for aerosol emission rates, the representation of aerosol size distributions, the quality and accuracy of input data, and the selection and implementation of parameterization schemes for various physical processes could all introduce uncertainties and potential impacts on the simulated results. Given this complexity, direct comparisons of the "Resolved" and "Interpolated" methods with observations may not provide a conclusive assessment of whether our modifications improve the model's overall

simulation abilities. Therefore, we didn't evaluate the model's simulation results of meteorological fields from two methods by comparing with more observation results other than the ERA5 reanalysis dataset. Our results show that the "Resolved" method can capture complex relationships between aerosols' optical properties and wavelengths that the "Interpolated" method may miss, particularly for dust-dominated regions and at specific wavelengths where water contents have a significant larger absorption than other wavelengths. Besides, our results demonstrate that the amendment of algorithms can significantly affect the simulation results of meteorological fields. While these changes may not necessarily lead to better agreement with observations in all cases, they give the model more potential to improve simulation abilities by more accurately representing the underlying physical processes.

**Code and data availability**

The current version of WRF-Chem is available from the project website:
http://www2.mmm.ucar.edu/wrf/users/download/get_source.html. The exact version of the model used to produce the results used in this paper is archived on Zenodo (https://doi.org/10.5281/zenodo.11244077 (Feng, 2024)), as are datasets and scripts to produce the plots for all the simulations presented in this paper. The model, datasets and scripts are under MIT licence.

**Author contributions**

Jiawang Feng and Chun Zhao developed the code. Jiawang Feng, Qiuyan Du, and Zining Yang conducted the experiments. Jiawang Feng and Chun Zhao analyzed the simulations. All authors contributed to the discussion and final version of the paper.

**Acknowledgments**

This research was supported by the National Key Research and Development Program of China (No. 2022YFC3700701), the Strategic Priority Research Program of Chinese Academy of Sciences (XDB0500303, XDB41000000), National Natural Science Foundation of China (41775146), the USTC Research Funds of the Double First-Class Initiative (YD2080002007, KY2080000114), the Science and Technology Innovation Project of Laoshan Laboratory (LSKJ202300305), and the National Key Scientific and Technological Infrastructure project "Earth System Numerical Simulation Facility" (EarthLab). The study used the computing resources from the Supercomputing Center of the University of Science and Technology of China (USTC) and the Qingdao Supercomputing and Big Data Center.

**Competing interests**

The authors declare that they have no conflict of interest.

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

**Table 1.** Lower/upper bound and volume fraction for eight size bins of aerosols in the model.

| Bin | Volume fraction | Lower bound of diameter ($\mu m$) | Upper bound of diameter ($\mu m$) |
|-----|-----------------|-----------------------------------|-----------------------------------|
| 1 | $1.43 \times 10^{-6}$ | 0.0390625 | 0.078125 |
| 2 | $2.71 \times 10^{-5}$ | 0.078125 | 0.15625 |
| 3 | $3.57 \times 10^{-4}$ | 0.15625 | 0.3125 |
| 4 | $3.34 \times 10^{-3}$ | 0.3125 | 0.625 |
| 5 | 0.022 | 0.625 | 1.25 |
| 6 | 0.1048 | 1.25 | 2.5 |
| 7 | 0.3425 | 2.5 | 5 |
| 8 | 0.4782 | 5 | 10 |


**Table 2**. Physical and chemical options of WRF-Chem used in this study.

| Model configuration | Description |
| --- | --- |
| Microphysics scheme | Morrison 2-moment (Morrison et al., 2009) |
| Short/Longwave radiation scheme | RRTMG (Mlawer et al., 1997; Iacono et al., 2000)s |
| Gas phase chemistry scheme | CBMZ (Zaveri and Peters, 1999) |
| Aerosol module | MOSAIC (Zaveri et al., 2008) |
| Boundary layer scheme | Yonsei University Scheme (YSU) (Hong et al., 2006) |
| Cumulus option | Kain-Fritsch Eta (Kain, 2004) |
| Land surface scheme | Noah (Chen and Dudhia, 2001) |

| Model configuration | Description |
| --- | --- |
| Microphysics scheme | Morrison 2-moment (Morrison et al., 2009) |
| Short/Longwave radiation scheme | RRTMG (Mlawer et al., 1997; Iacono et al., 2000)s |
| Gas phase chemistry scheme | CBMZ (Zaveri and Peters, 1999) |
| Aerosol module | MOSAIC (Zaveri et al., 2008) |
| Boundary layer scheme | Yonsei University Scheme (YSU) (Hong et al., 2006) |
| Cumulus option | Kain-Fritsch Eta (Kain, 2004) |
| Land surface scheme | Noah (Chen and Dudhia, 2001) |

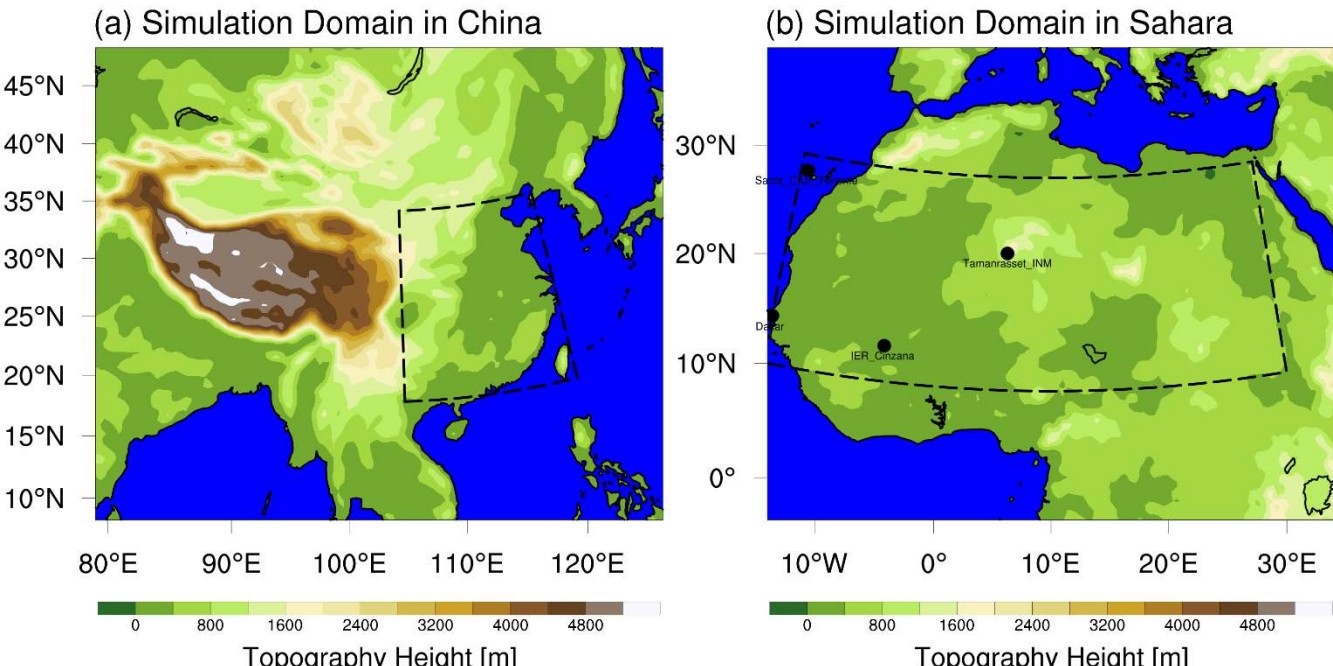

**Figure 1.** Simulation domains. **(a)** in China; **(b)** in Sahara. The dashed-line boxes in panel **(a)** and **(b)** represent regions dominated by anthropogenic aerosols and dust aerosols, respectively. Spatial distributions of topography height are also shown. Locations of selected AERONET stations over dust-dominant areas in Sahara are also denoted in panel (b).

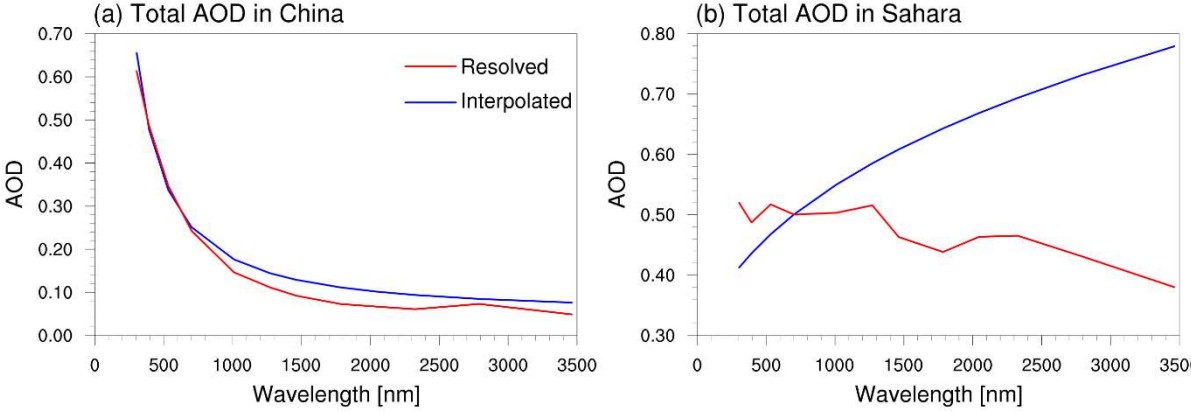


**Figure 2.** Simulated AOD as a function of wavelength, averaged for January and July 2015 **(a)** AOD averaged over anthrodominant region in East China (as shown in Fig. 1); **(b)** AOD averaged over dust-dominant region in the Sahara (as shown in Fig. 1). The blue and red line represent the "Interpolated" and the "Resolved" method, respectively.

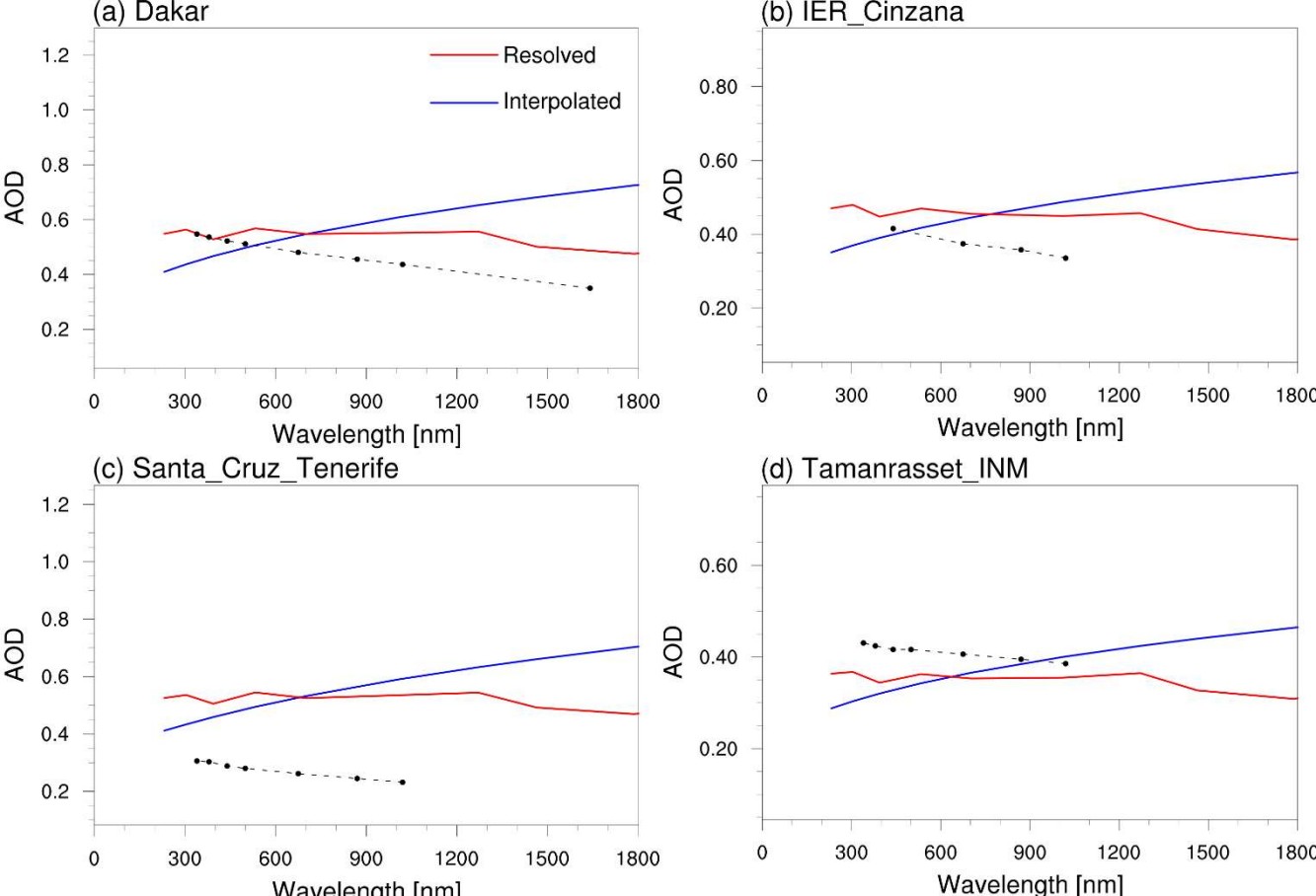

**Figure 3.** Comparison of total AOD from simulations and AERONET observations in dust-dominant areas. The results are both averaged for January and July 2015. The blue and red line represent the "Interpolated" and the "Resolved" method, respectively. The AERONET AOD values are indicated by black dots in each panel. The simulation results are obtained 680 from the grid box closest to the AERONET stations.

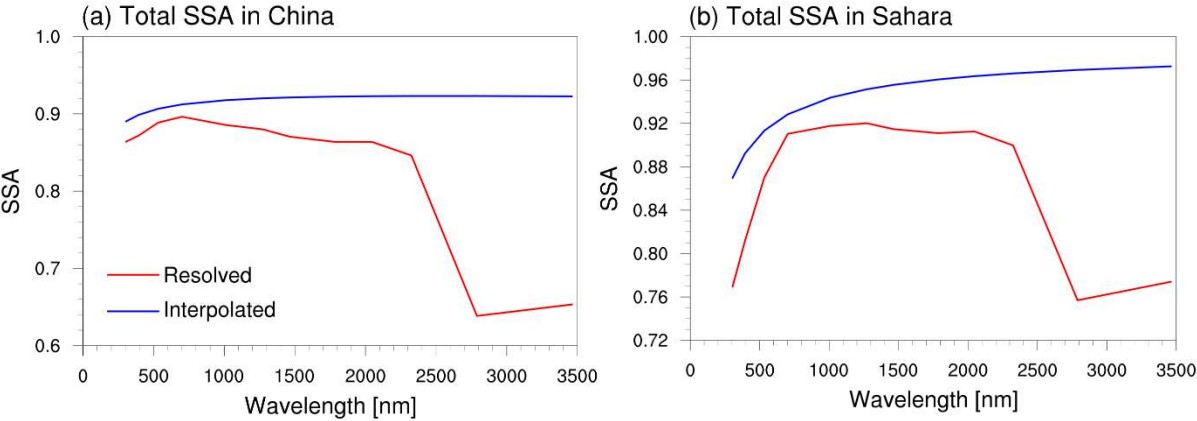

**Figure 4.** Simulated SSA as a function of wavelength, averaged for January and July 2015 (a) SSA averaged over anthro-dominant region in East China (as shown in Fig. 1); (b) SSA averaged over dust-dominant region in the Sahara (as shown in Fig. 1). The blue and red line represent the "Interpolated" and the "Resolved" method, respectively.


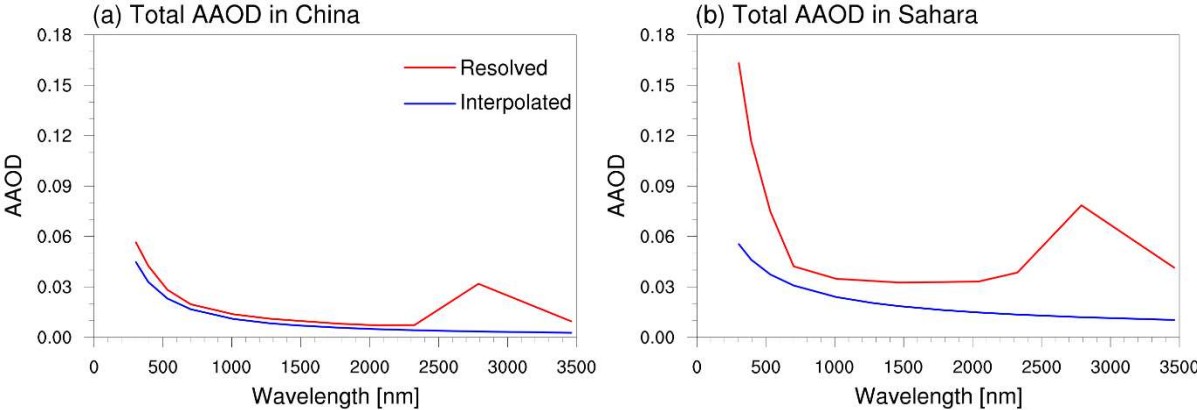

**Figure 5.** Simulated AAOD as a function of wavelength, averaged for January and July 2015 (a) AAOD averaged over
anthro-dominant region in East China (as shown in Fig. 1); (b) AAOD averaged over dust-dominant region in the Sahara (as
shown in Fig. 1). The blue and red line represent the "Interpolated" and the "Resolved" method, respectively.

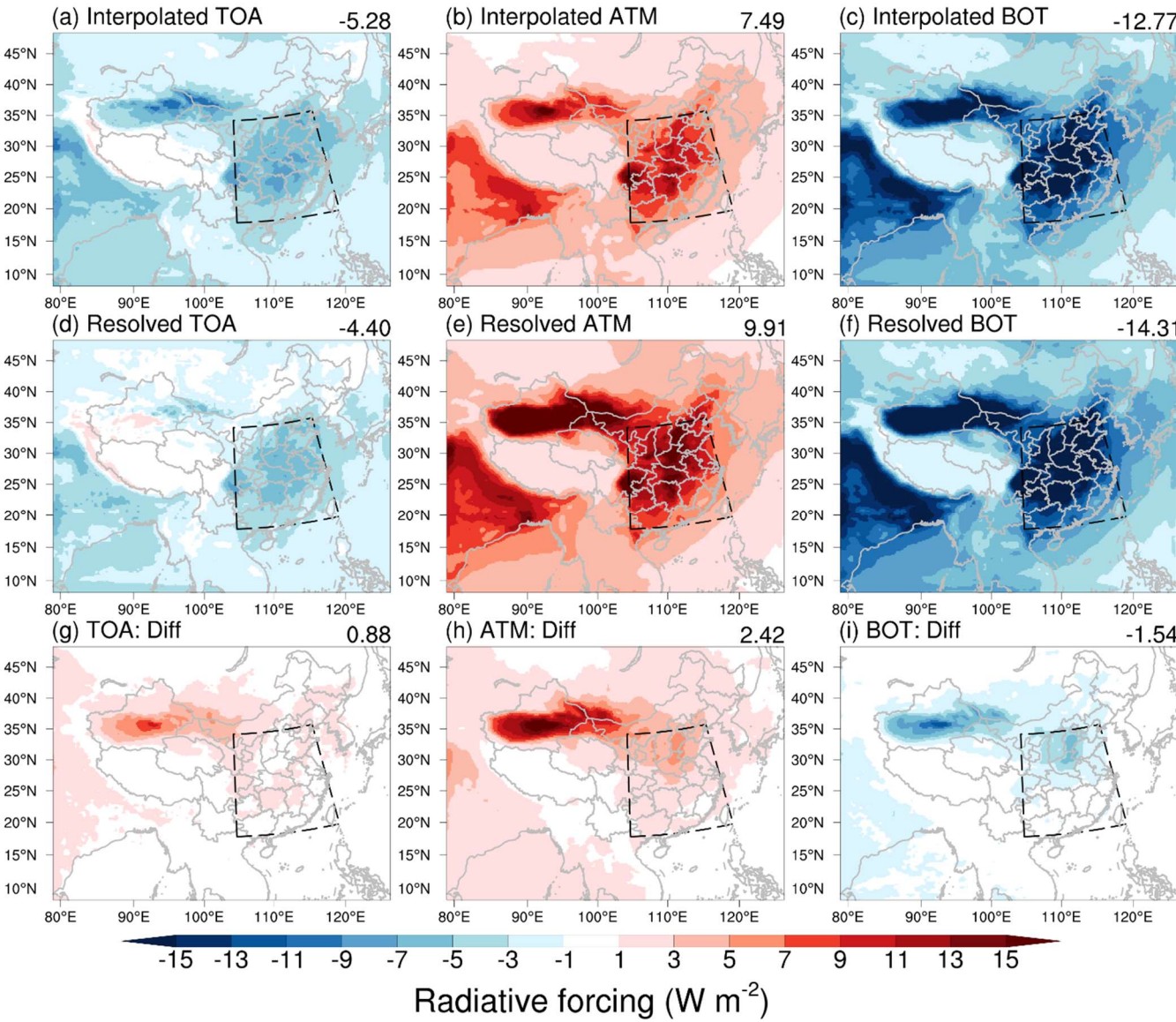

ss

**Figure 6.** Radiative forcing of all aerosols in China at TOA, BOT, and in ATM, averaged for January and July 2015. The top and middle panels show the results using "Interpolated" and "Resolved" methods, respectively. The bottom panels show the differences between the "Interpolated" method and the "Resolved" method. The average results of anthro-dominant areas (dashed-line boxes) are shown in the top right corner.


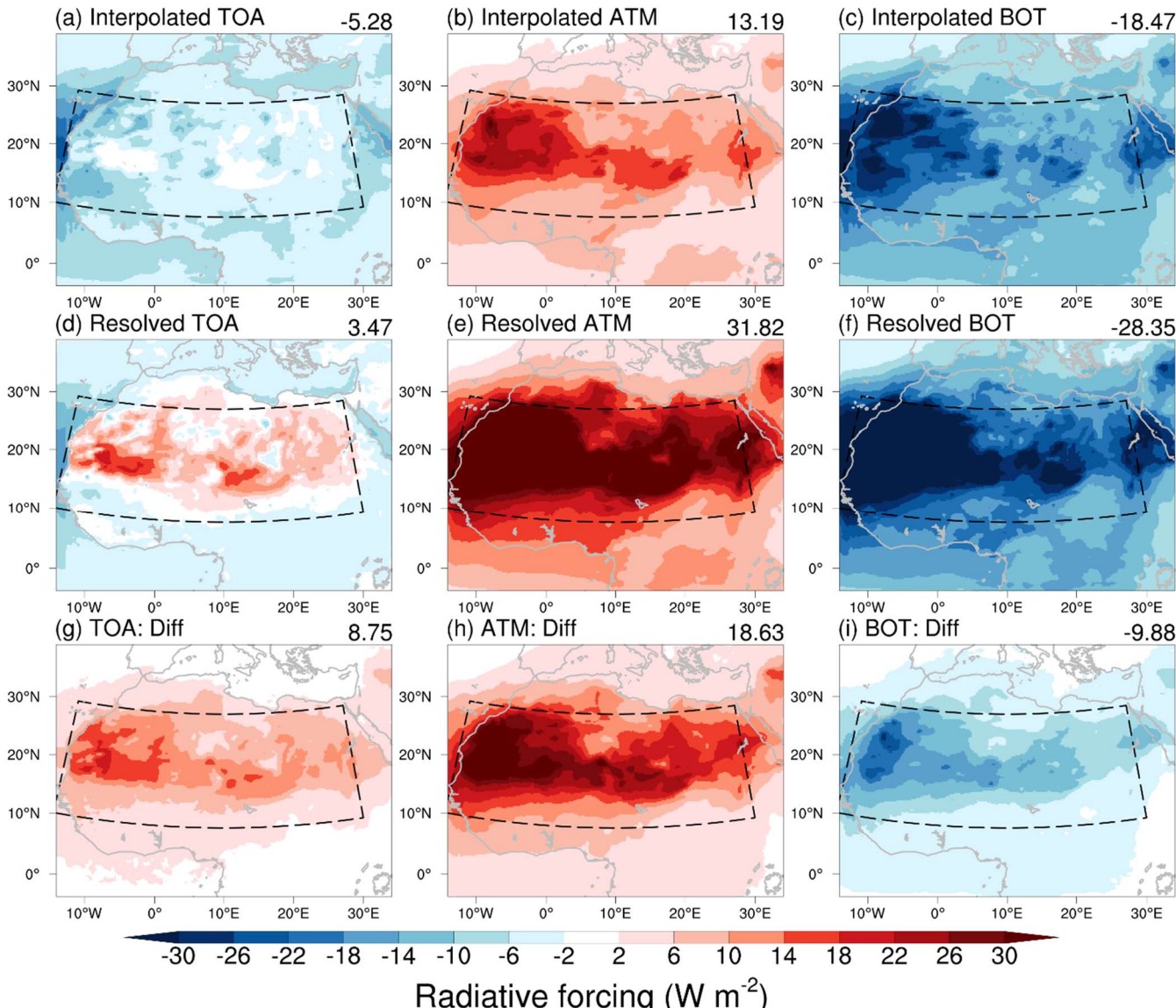

**Figure 7.** Radiative forcing of aerosols in Sahara at TOA, BOT, and in ATM, averaged for January and July 2015. The top and middle panels show the results using "Interpolated" and "Resolved" methods, respectively. The bottom panels show the differences between the "Interpolated" method and the "Resolved" method. The average results of dust-dominant areas (dashed-line boxes) are shown in the top right corner.

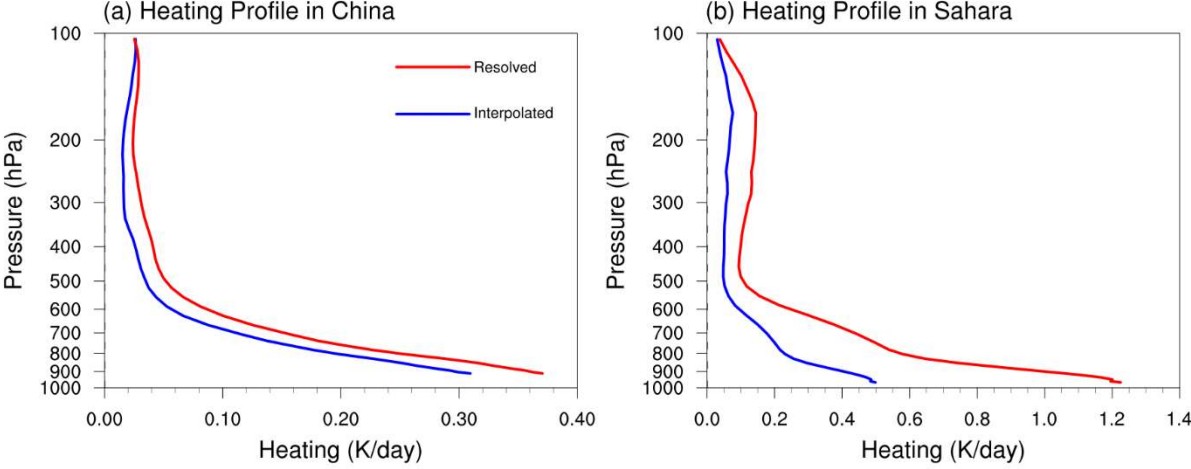

**Figure 8.** Shortwave heating profile of aerosols, averaged for January and July 2015. **(a)** over anthro-dominant areas in China; **(b)** over dust-dominant areas in Sahara.


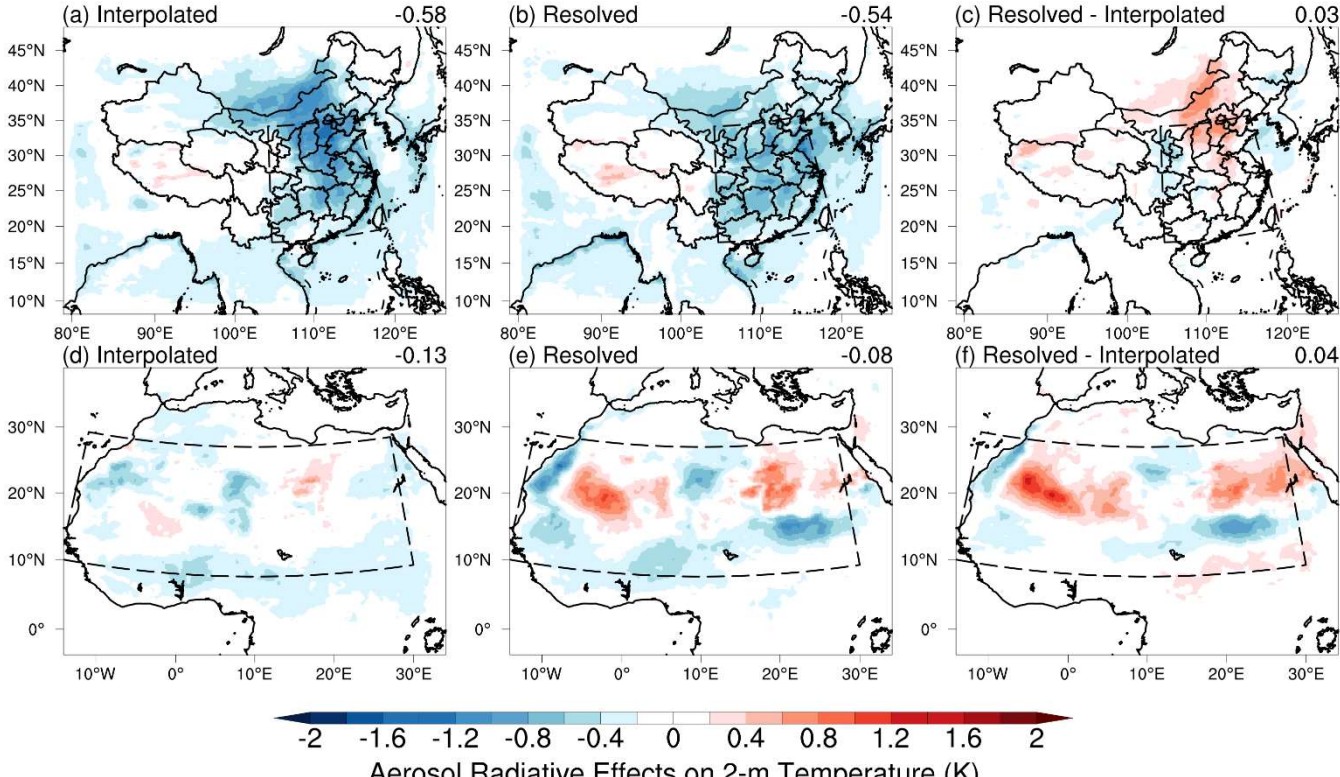

**Figure 9. (a, d)** Spatial distribution of aerosol radiative effects on 2-m temperature from the "Interpolated" experiments in China and Sahara, respectively; **(b, e)** Spatial distribution of aerosol radiative effects on 2-m temperature from the "Resolved" experiments in China and Sahara, respectively; **(c, f)** The impacts of algorithm amendment on the simulated aerosol radiative effects (difference in aerosol radiative effects between "Resolved" and "Interpolated") on 2-m temperature in China and Sahara, respectively. The average results of anthro-dominant (dashed-line boxes in a, b, and c) and dust-dominant areas (dashed-line boxes in d, e, and f) are shown in the top right corner. The results are averaged for January and July 2015.


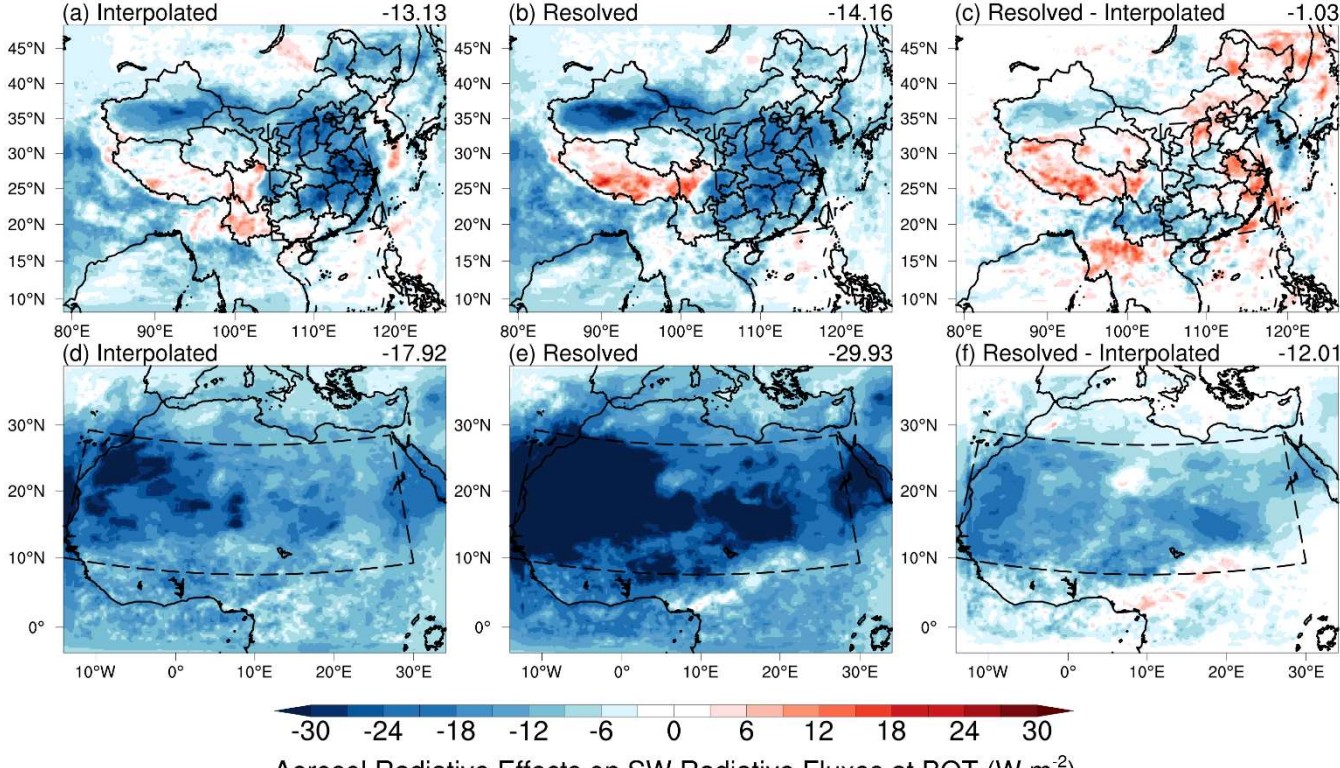

**Figure 10.** Same as Fig. 9 but for the net short-wave radiative fluxes at the bottom of the atmosphere (positive denotes downward).

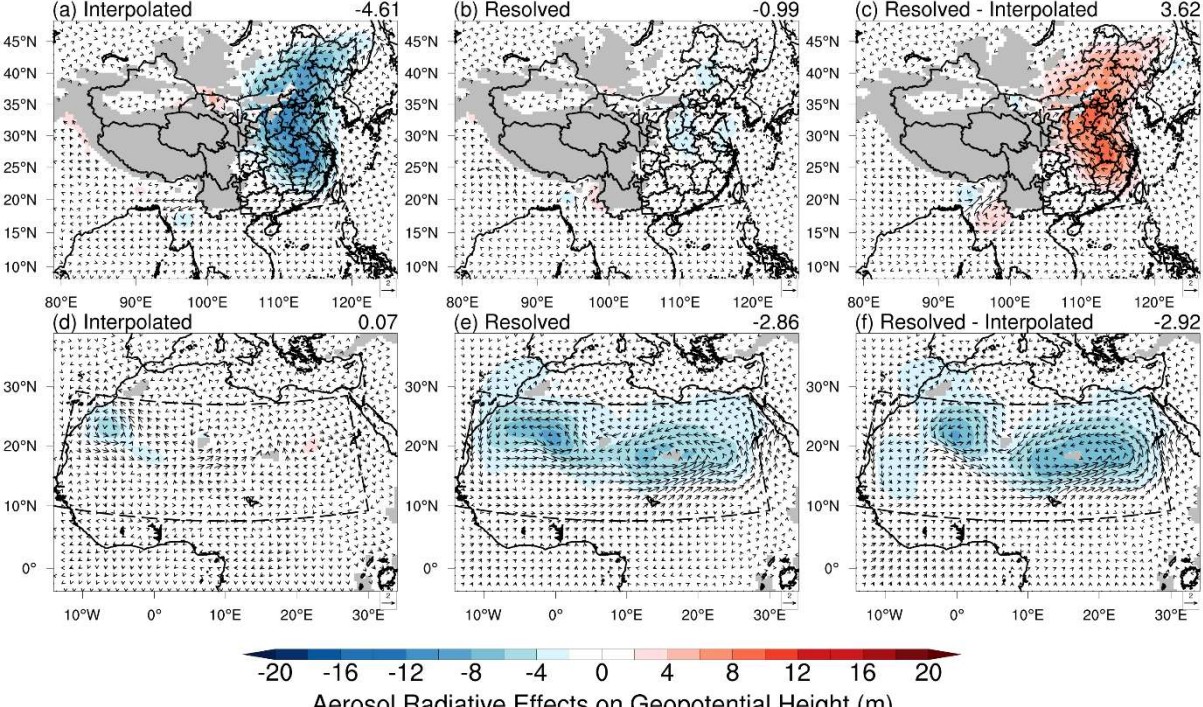


**Figure 11.** Same as Fig. 9 but for wind fields (vectors) and geopotential height (shaded contour) at 850 hPa.

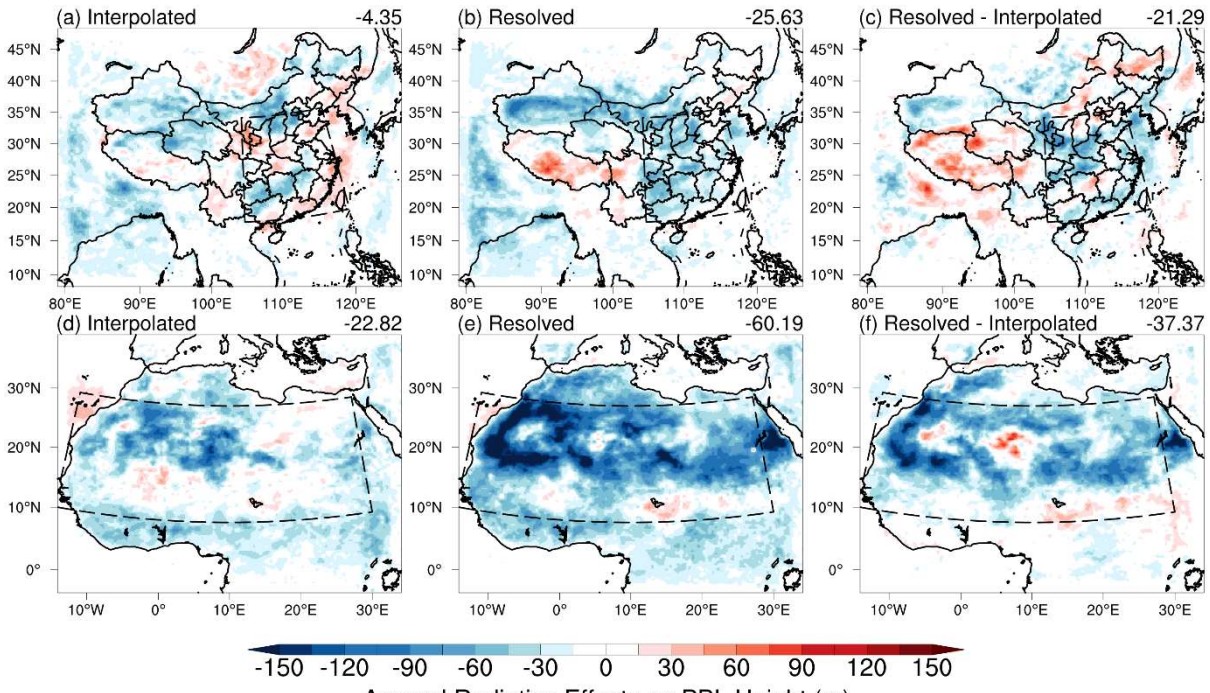

**Figure 12.** Same as Fig. 9 but for PBL height.


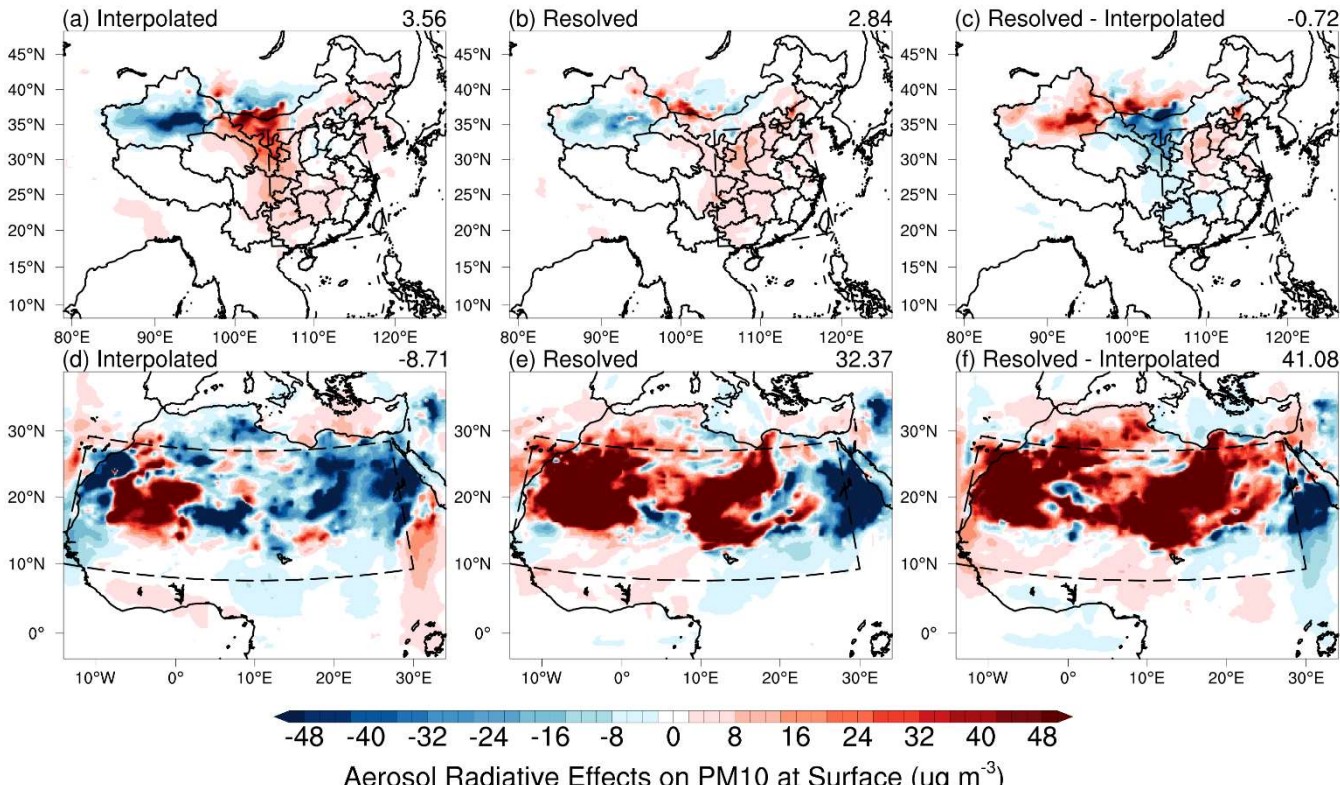

**Figure 13.** Same as Fig. 9 but for particulate matter with diameters 10 μm and smaller (PM10) at surface.