# Peer review of "Amending the algorithm of aerosol-radiation interaction in WRF-Chem (v4.4)"

_Geoscientific Model Development, 2024_

## Author Comment (AC1)

**Response to Referee #1**

*This paper describes how the aerosol-radiation calculations in the WRF-chem model are modified to include more wavelengths that match with input needed by the model's radiation parameterization. The subject matter is appropriate for GMD. While the paper is generally well written, easy to follow, and contains important information for users interested in aerosol-radiation interactions, there are several issues that need to be resolved before the paper is suitable for publication.*

**Response:** We thank you for your time and expertise. Your constructive criticism has been instrumental in refining our work. We have carefully addressed each of your specific comments, as detailed in our point-by-point responses below. We believe these revisions have enhanced the manuscript and hope that they adequately address your concerns.

*General comments:*

- *Context: The paper demonstrates that differences between the "interpolated" and "resolved" simulations of aerosol optical properties and consequently aerosol radiative forcing can be large, usually for areas dominated by dust. However, they fail to mention other assumptions in the model which could impact those results. Since the effects of dust in their "resolved" simulation seem to be related to absorption, model predictions will depend significantly on refractive indices used for both BC and dust. Presumably dust in this model is assumed to be absorbing to some extent. Yet, there are many studies measuring refractive indices of dust throughout the world, showing large difference resulting from different mineral compositions that vary from region to region. Therefore, this parameter does not seem to be well constrained. If one was to replace the refractive indices with less absorbing dust, the differences between the "interpolated" and "resolved" simulations would presumably be smaller. In addition, the aerosol size distribution will have an impact on the aerosol optical properties, but the paper does not provide any information on how the size distribution of dust is represented. These are complicated issues which are coupled to model evaluation (see next comment). At a minimum the authors need to include discussion somewhere in the text and the conclusions to put their results into the proper context.*

**Response:** We appreciate your insightful comments regarding the potential impact of model assumptions on our results, particularly concerning aerosols optical properties. We acknowledge that these are indeed important considerations, and we would like to address them as follows:

Refractive indices and regional variability of dust compositions: We agree that the refractive indices of dust can vary significantly depending on mineral composition and source region. Our study uses a set of refractive indices that represent a global average, which may not capture the full range of variability observed in different regions. While

changing these values could indeed affect the magnitude of the differences between the "Interpolated" and "Resolved" simulations, it's important to note that the main objective of our study is to demonstrate the necessity of multi-band resolution in representing aerosol optical properties, rather than to provide the most accurate simulated fields, such as total AOD and radiative forcing, that, as the reviewer pointed, can be affected by many model parameters and processes. We have conducted more experiments with smaller radiative indices for dust aerosols. The results approved your opinion that with less absorbing dust, the differences of radiative effects could become smaller. However, the spectral variation of optical properties would still be better captured by the "Resolved" method compared to the "Interpolated" method. We have added a detailed discussion in revised manuscript about how "Resolved" method better captured the spectral variation of optical properties: "AERONET observations demonstrate a gradual decline in dust Aerosol Optical Depth (AOD) with increasing wavelength, with minimal variation between 400 nm and 600 nm. Although both the "Resolved" and "Interpolated" algorithms incorrectly predict an increasing trend in AOD between 400 nm and 600 nm, the "Resolved" method's more detailed spectral optical calculations offer the ability of correcting this discrepancy at other wavelengths. Conversely, the "Interpolated" algorithm's erroneous trend at these wavelengths, when applied in the Ångström exponent theory, propagates and amplifies discrepancies at longer wavelengths. These findings underscore the need for cautious application of the Ångström exponent theory, particularly when dealing with aerosols such as dust, whose optical properties exhibit minimal variation at the selected wavelengths (400 nm and 600 nm in WRF-Chem). In such cases, small uncertainties can lead to a reversal in the AOD-wavelength relationship, potentially resulting in significant simulating errors in spectral extrapolation."

BC aerosol: Actually, the refractive indices of BC aerosols are the same (1.95 + 0.79i) in both original WRF-Chem ("Interpolated" method) and OPAC dataset (used in "Resolved" method). Therefore, the absorption effects of BC aerosols are similar between these two methods. To demonstrate the effects of choice of refractive indices, we conducted further experiments using refractive indices of BC in Flanner et al. (2012). The AOD and AAOD results in China are listed below:

[Figure]

The results are still similar to using original WRF-Chem and OPAC dataset. The reason may be that BC aerosols are more aligned to the Ångström law compared to dust aerosols. The peak of AAOD at around 2900nm are discussed later in this response. We add discussions in revised manuscript: "Additionally, the simulated total AOD was also

compared with the AERONET results in anthro-dominant regions of China (see Supplementary Fig. S1). The results demonstrate that both the "Interpolated" and "Resolved" algorithm simulations exhibit significant decreasing trends in total AOD as those seen in the AERONET data. Therefore, the discrepancies introduced by the application of the Ångström exponent theory has much smaller effects on simulating results, which further indicates that the amendments to the algorithm have less impact on simulated AOD resulting from anthropogenic aerosols compared to dust aerosols."

Size distribution representation: Thanks for this great suggestion. Now, we add more information about the size distribution of aerosols in our model in Table 1 of revised manuscript: "the size distribution of emitted dust particles follows a theoretical expression based on the physics of scale-invariant fragmentation of brittle materials derived by Kok (2011). The detailed aerosols size distribution from approximately 0.04 μm to 10 μm are listed in Table 1."

**Table 1.** Lower/upper bound and volume fraction for eight size bins of aerosols in the model.

| Bin | Volume fraction | Lower bound of diameter ($\mu m$) | Upper bound of diameter ($\mu m$) |
|---|---|---|---|
| 1 | $1.43 \times 10^{-6}$ | 0.0390625 | 0.078125 |
| 2 | $2.71 \times 10^{-5}$ | 0.078125 | 0.15625 |
| 3 | $3.57 \times 10^{-4}$ | 0.15625 | 0.3125 |
| 4 | $3.34 \times 10^{-3}$ | 0.3125 | 0.625 |
| 5 | 0.022 | 0.625 | 1.25 |
| 6 | 0.1048 | 1.25 | 2.5 |
| 7 | 0.3425 | 2.5 | 5 |
| 8 | 0.4782 | 5 | 10 |

Context and implications: We appreciate your suggestion to provide more context for our results. We have added a discussion in the manuscript acknowledging these limitations and uncertainties. Theoretically, one could identify a set of RI that would yield similar aerosol radiative forcing results for the "Resolved" and "Interpolated" algorithms. However, such a set might not accurately reflect the true physical properties of the aerosols. Despite these considerations, it is important to note that the spectral variation of optical properties would be better captured by the "Resolved" method compared to the "Interpolated" method as discussed above and in the revised version of Section 3.2. This provides a more robust foundation for accurately representing the complex interactions between aerosols and radiation across various wavelengths. The advantages of the "Resolved" method in capturing these spectral variations persist regardless of the specific refractive indices employed in the simulations.   In the revised manuscript, we add: "It should be noted that: the refractive indices of dust can vary significantly depending on mineral composition and source region. Our study uses a set of refractive indices that represent a global average, which may not capture the full range of variability observed in different regions. Changing these values could affect

the magnitude of the radiative forcing differences between the "Interpolated" and "Resolved" simulations. We have conducted additional sets of experiments with smaller radiative indices for dust aerosols in "Resolved" simulations. The results show that with less absorbing dust (smaller imaginary part of refractive indices), the differences of radiative effects could become smaller (not shown here). Theoretically, one could identify a set of aerosol optical properties that would yield similar aerosol radiative forcing results for the "Resolved" and "Interpolated" algorithms. However, such a set might not accurately reflect the true physical properties of the aerosols. Despite these considerations, it is important to note that the spectral variation of optical properties would be better captured by the "Resolved" method compared to the "Interpolated" method as discussed in Section 3.2. This provides a more robust foundation for accurately representing the complex interactions between aerosols and radiation across various wavelengths. The advantages of the "Resolved" method in capturing these spectral variations persist regardless of the specific refractive indices employed in the simulations."

- *Physical Interpretation: The paper focuses on differences between "interpolated" and "resolved" simulations, but there is very little investigation into the physical reasons for the differences (other than absorption in general) and explain that to the reader. A little more discussion is needed. For example, the drop in SSA around 2500 corresponds to an increase in absorption. Is the increase in absorption due to larger concentrations of coarse aerosols? How do we know whether the model not just overestimating dust? While that might explain the different trends in SSA at longer wavelengths, it could impact the magnitude of the differences. Or are there other factors?*

**Response:** Thank you for your insightful suggestions. The sudden decrease in SSA around 2900nm is primarily due to the internal mixing of water aerosol. In WRF-Chem, the optical calculations include various aerosol species and their water content. Water exhibits a peak in absorption at approximately 2900nm, which significantly influences the overall aerosol optical properties at this wavelength. We have added more explanations in the revised manuscript, such as the differences patterns in SSA and AAOD between the two methods: "In WRF-Chem, water content of aerosols is also considered while calculating optical properties. Water exhibits a peak in absorption at approximately 2900nm, which significantly influences the overall aerosol optical properties at this wavelength. As illustrated in Figure S2 of the supplementary materials, there is a pronounced peak in the imaginary part of the refractive index of water at around 2900nm in "Resolved" method. The imaginary part of the refractive index is directly related to absorption, with higher values indicating stronger absorption. This peak in water absorption leads to the observed increase in AAOD (decrease in SSA) at this wavelength. In contrast, the "Interpolated" method, which relies solely on optical property information at 400 nm and 600 nm, fails to capture this crucial spectral feature. Consequently, the "Interpolated" method is unable to accurately represent the complex wavelength-dependent optical properties of aerosols. Besides, the lower SSA values and higher AAOD values obtained from the "Resolved" method compared to the

"Interpolated" method in the Sahara region are primarily due to the spectral variation of dust optical properties, particularly at shorter wavelengths. In OPAC dataset, dust has a larger imaginary part of the refractive index at wavelengths shorter than ~600nm, indicating stronger absorption in this spectral range (also as shown in Figure S2). However, in the latest version of WRF-Chem ("Interpolated" method), the imaginary part of the refractive index for dust is set to a constant value of 0.003 at all four shortwave bands. The "Resolved" method simulates higher AAOD by using the wavelength-dependent refractive indices, especially at shorter wavelengths. This results in significant differences in SSA and AAOD between the two methods, particularly in dust-dominated regions like the Sahara."

[Figure]

**Figure S2.** Imaginary part of refractive index used in "Resolved" method. The blue and red lines represent water and dust, respectively.

- *Evaluation: Most of the figures show differences between the "interpolated" and "resolved" aerosol optical properties simulations. While there are a few instances of comparing results with observations (e.g. AERONET in Fig 3, ERA5 in Supplemental material), there seems to be a missed opportunity to use more observations to convince the reader that the "resolved" method is indeed better. For example, why not show comparisons with AERONET in the China domain? Why not compare TOA forcing to CERES? These comparisons are relatively easy to do. There are also many observations of scattering and absorption that could have been used to quantify model performance and better understand how well absorption is handled by the new treatment. Meteorological comparisons use ERA5, but ERA5 is a proxy for observations and might have larger uncertainties where in situ measurements are sparse, but the authors do not comment such caveats. In summary, I would encourage the authors to do a more thorough job of evaluation.*

**Response:** We appreciate your suggestion to include more observational comparisons in our study. While we understand the value of such comparisons, we would like to address this concern and explain our approach:

Focus of the study: The primary aim of our study is to demonstrate the importance of using a spectrally resolved method for calculating aerosol optical properties, rather than to improve the overall model performance. We show that the "Resolved" method can capture complex relationships between aerosols' optical properties and wavelengths that the "Interpolated" method may miss, particularly for dust-dominated regions and at specific wavelengths. Besides, our results demonstrate that the amendment of algorithms can significantly affect the simulation results. While these changes may not necessarily lead to better agreement with observations in all cases, they give the model more potential to improve simulation abilities by more accurately representing the underlying physical processes.

Complexity of model evaluation: It's important to note that many factors can affect the simulation of meteorological fields and aerosol properties beyond the optical properties methods we're investigating such as model resolution, physical algorithms, parameterization schemes, and input data quality. Given this complexity, direct comparisons of the "Resolved" and "Interpolated" methods with observations of meteorological fields may not provide a conclusive assessment of whether our modifications improve the model's overall simulation abilities. Nonetheless, we tried our best to provide evaluations on the trend of aerosols' optical properties with wavelengths and the dust direct radiative forcing. We add some further discussions about dust radiative forcing at TOA in revised manuscript: "Furthermore, our previous study (Feng et al., 2023) initially employed the original optical calculation algorithm in WRF-Chem (referred to as the "Interpolated" algorithm), which yielded a dust direct radiative forcing at the top of the atmosphere (TOA) of -0.75 W/m². This value significantly deviated from the observationally constrained estimate of -0.20 W/m² proposed by Kok et al. (2017). Subsequently, in the final version of Feng et al. (2023), we implemented the "Resolved" method, which resulted in a substantially improved estimate of -0.27 W/m². This marked improvement in alignment with observational constraints strongly suggests that the "Resolved" method demonstrates better performance in simulating radiative forcing, particularly for dust aerosols."

Limitations of observational comparisons: We acknowledge that additional comparisons with observations like CERES for TOA forcing could provide valuable insights. However, interpreting these comparisons would require careful consideration of other model uncertainties. The accurate simulation of TOA forcing is contingent upon numerous model parameters and processes, including but not limited to: the calibration factors for aerosol emission rates, the representation of aerosol size distributions, and the parameterization schemes employed for deposition processes. These factors are subject to substantial uncertainties in current modelling frameworks and can significantly impact the spatial distribution of aerosols, which in turn directly affects the simulated TOA forcing. Observational datasets like CERES provide integrated measurements of TOA forcing, encompassing the cumulative effects of all these processes. Consequently, isolating the specific impact of our amendments of

optical properties calculation algorithm from these other sources of uncertainties presents a significant challenge, which is beyond the scope of our current study.

We have added a caveat in the manuscript acknowledging our limitations and adding more discussions as: "It's important to note that the primary aim of our study is to demonstrate the importance of using a spectrally resolved method for calculating aerosol optical properties, rather than to improve the overall model performance. Many factors can affect the simulation of meteorological fields and radiative processes beyond the optical properties methods we're investigating in this study. For example, while our study employs a 50km grid resolution, which is suitable for investigating aerosol-radiation interactions, higher-resolution could enhance the simulation of aerosol emission, deposition, and transport processes, potentially leading to a more accurate representation of aerosol distributions and their radiative effects (Feng et, al., 2023; Tan et, al., 2015; Tao et, al., 2020). In addition to model resolution, the calibration factors for aerosol emission rates, the representation of aerosol size distributions, the quality and accuracy of input data, and the selection and implementation of parameterization schemes for various physical processes could all introduce uncertainties and potential impacts on the simulated results. Given this complexity, direct comparisons of the "Resolved" and "Interpolated" methods with observations may not provide a conclusive assessment of whether our modifications improve the model's overall simulation abilities. Therefore, we didn't evaluate the model's simulation results of meteorological fields from two methods by comparing with more observation results other than the ERA5 reanalysis dataset. Our results show that the "Resolved" method can capture complex relationships between aerosols' optical properties and wavelengths that the "Interpolated" method may miss, particularly for dust-dominated regions and at specific wavelengths where water contents have a significant larger absorption than other wavelengths. Besides, our results demonstrate that the amendment of algorithms can significantly affect the simulation results of meteorological fields. While these changes may not necessarily lead to better agreement with observations in all cases, they give the model more potential to improve simulation abilities by more accurately representing the underlying physical processes."

- *Figures: The authors need to be more explicit in what their figures are showing. I assume that most of them are monthly averages, but the figure captions and text need to make that clear.*

**Response:** Thanks for your comment. Sorry for the confusion and yes, most of the results are monthly averages. Now, we' add further descriptions in the figure captions and text. In the revised manuscript, we add: "In this study, all analysis results, unless otherwise stated, represent monthly averages for January and July 2015" Specifically: in figures 2~13 captions, we add the phrases like this: "averaged for January and July 2015"

- *Computational Aspects: The authors should describe the computational costs of computing the aerosol optical properties for the "revolved" vs "interpolated"*

*method. Presumably it is 3.5 times more, but they should be explicit. Would also be useful to put those costs in perspective with the overall runtime of their simulations. Somewhere in the paper (Section 2.3?), they should also note how often the aerosol optical properties are updated. In addition, the aerosol optical property calculations in WRF-Chem work for bulk, modal, and sectional representations of aerosol. Do the modifications follow that format, or do they only work with the sectional representation used in this study? While I do not think it is necessary to compare similar differences between the "interpolated" and "resolved" methods for bulk and modal aerosol representations it would be useful to let the reader know about these details.*

**Response:** We appreciate your suggestion to provide more details about the computational costs and implementation of our methods. We have added the following information to Section 2.3 of our revised manuscript: "We have conducted a detailed analysis of the computational costs associated with both the 'resolved' and 'interpolated' methods. For the 'resolved' method, the calculation of aerosol optical properties takes 11,458.6 seconds, which accounts for 9.3% of the total simulation runtime of 122,717 seconds. In contrast, the 'interpolated' method requires 5,223.05 seconds for the same calculations, which accounts for 4.9% of its total runtime of 107,615 seconds. Therefore, the "Resolved" method takes approximately 2.19 times computational cost for aerosol optical property calculations compared to the "Interpolated" method. This difference in calculation time translates to an additional 14% in total simulation runtime when using the 'resolved' method. It's worth noting that this increase in computational cost is less than the 3.5 times one might expect from increasing the number of shortwave bands from 4 to 14. This is because the aerosol optical properties process includes both shortwave and longwave calculations. The original WRF-Chem has already used "Resolved" method to calculate the longwave part, therefore, we only modified the shortwave calculations from 4 bands to 14 bands." And "The model time step is set at 150 seconds, and the aerosol optical properties are updated every 30minutes in the model."

Furthermore, we have clarified in Section 2.2 that "While our study focused on the sectional representation of aerosols, our amendment of algorithm is valid for bulk, sectional and modal representations in WRF-Chem, theoretically. However, we have not tested the differences between "Interpolated" and "Resolved" methods for bulk and modal aerosol representations in this study."

*Specific Comments:*
- *Lines 37-40: These sentences were generally true several years ago, but many global models now can be run in a regional-refined mode similar to WRF or run globally at kilometer scale grid spacings. While they may not have as sophisticated chemistry, they do have treatments of aerosols and aerosol-radiation interactions.*

**Response:** We sincerely appreciate your careful review and for bringing this to our attention. We acknowledge that our original discussion lacked precision and rigor. We have revised the relevant sentences in the manuscript to more accurately reflect the capabilities of WRF-Chem. The updated text now reads: "WRF-Chem is capable of performing regional-scale simulations with high spatial resolution. This allows for a detailed representation of aerosol and radiation processes at regional scale."

- **Line 69: Would be useful to include a table of physics parameterizations used in WRF, so users to not have to search for those in another paper.**

**Response:** Thank you for your suggestion. We have added a table of physics parameterizations used in WRF in Section 2.3:

The detailed parameterization schemes of physical and chemical processes of the WRF-Chem model used in the study are summarized in Table 2.

Table 2. Physical and chemical options of WRF-Chem used in this study.

| Model configuration | Description |
| --- | --- |
| Microphysics scheme | Morrison 2-moment (Morrison et al., 2009) |
| Short/Longwave radiation scheme | RRTMG (Mlawer et al., 1997; Iacono et al., 2000)s |
| Gas phase chemistry scheme | CBMZ (Zaveri and Peters, 1999) |
| Aerosol module | MOSAIC (Zaveri et al., 2008) |
| Boundary layer scheme | Yonsei University Scheme (YSU) (Hong et al., 2006) |
| Cumulus option | Kain-Fritsch Eta (Kain, 2004) |
| Land surface scheme | Noah (Chen and Dudhia, 2001) |

- **Lines 88-89: Would be useful to number the equations. Also, this method implies that AOD and forcing is linear. Is that true?**

**Response:** We appreciate your suggestion and insightful question about the linearity of AOD and forcing. We have numbered the equations in the revised manuscript as suggested.

Regarding the question of linearity, the aerosols in WRF-Chem are assumed to be internally mixed, therefore, he AOD and forcing are not linearity. We referenced the methods in Zhao et, al., (2013), in which the calculation of aerosol optical properties and radiative transfer is performed multiple times with the mass of one or more aerosol species and also its associated water aerosol mass removed from the calculation each time. Helping us to diagnose AOD and forcing of different kinds of aerosols even they are not linear.

- **Lines 135-142: It would be useful to include the AERONET site locations used in this study in Figure 1, rather than in S1. Is there a reason other datasets, like**

*MODIS satellite AOD, are not used? It might be useful to explain to the reader that AERONET data is available at multiple wavelengths, but that may not be the case for other datasets. I assume this is one reason, but there could be others.*

**Response:** Thanks for your suggestion. We have addressed these points as follows:

Figure placement: We have moved the AERONET site locations from Figure S1 to Figure 1 in the main text as suggested. This change will make it easier for readers to reference the observation locations throughout the paper.

Use of other datasets: We thank the reviewer for inquiring about the use of other datasets, particularly MODIS satellite AOD. There are two main reasons why we chose to focus on AERONET data: a) Spectral information: AERONET provides AOD measurements at multiple wavelengths, which is crucial for our study comparing the 'Interpolated' and 'Resolved' methods across different wavelengths. b) Comparison relevance: MODIS satellite AOD is typically available only at 550 nm. For the 'Interpolated' and 'Resolved' methods, the AOD results are similar at 550 nm because both methods use the same calculation approach when the wavelength is the same. While satellite results could cover more regions, the comparison at a single wavelength would not effectively demonstrate the differences between our methods. The multi-wavelength data from AERONET allows us to show these differences more clearly. In the revised manuscript, we add these discussions: "To evaluate the modelling results, multi-spectral aerosol optical depth (AOD) measurements are required, which is critical for this study comparing the 'Interpolated' and 'Resolved' methods across wavelengths. several datasets are used in this study. The Therefore, we retrieved total AOD is from the AERONET network (Holben et al., 1998). Although satellite AOD products such as from the Moderate Resolution Imaging Spectroradiometer (MODIS) have greater spatial coverage, the number of wavelengths is limited. Comparison at only one or very few wavelengths would not effectively demonstrate the distinctions between the two algorithms examined in this work."

- *Line 144-146: Somewhere you should mention that this result is averaged over space and time for the entire domain from the simulation.*

**Response:** Thank you for your suggestion and sorry for the confusion. We add this description in the revised manuscript as: "The results presented are spatial averages over the regions delineated by the dashed boxes in Figure 1, as well as temporal averages for the months of January and July 2015."

- *Line 148-151: This sentence has a lot of commas, so the thoughts are difficult to follow. Suggesting breaking this up into at least 2 sentences so that the message is clearer.*

**Response:** Thanks for your suggestion, we revised this sentence as: "As seen in Figure 2a, both the 'Resolved' and 'Interpolated' algorithms produce similar exponential decaying trends in AOD for regions dominated by anthropogenic aerosols. The AOD values calculated with the 'Resolved' algorithm are slightly higher than those obtained with the 'Interpolated' algorithm. These trends indicate that the Ångström's theory is applicable to a certain extent in these areas."

- *Figure 3: is it necessary to go out to 3500 nm when the observations stop at 1500 nm? I believe AERONET does have observations at these high wavelengths, but not for these stations? Are the very long wavelengths important for the radiation calculations? If there is a reason to plot the long wavelengths, the text needs to describe why.*

**Response:** Thanks for your comment and suggestion. Actually, there is no specific reason to go out to 3500 nm. We revised the figures as suggested in the revised version. Now the range of wavelengths in Figure 3 is 0~1800nm.

- *Line 164: It would be useful to include a figure similar to figure 3, but for the China domain. Rather than showing results at individual sites, one could average the results over multiple sites.*

**Response:** Thank you for your great suggestion. We add a figure in the supplement material similar to figure 3 for the China domain. And add a discussion in the text: "Additionally, the simulated total AOD was also compared with the AERONET results in anthro-dominant regions of China (see Supplementary Fig. S1). The results demonstrate that both the "Interpolated" and "Resolved" algorithm simulations exhibit similar decreasing trends in total AOD as those seen in the AERONET data. This further indicates that the amendments to the algorithm have less impact on simulated AOD resulting from anthropogenic aerosols compared to dust aerosols."

As for the suggestion of averaging the results over sites, we found it can be confusing for readers since the valid ranges of wavelength for AERONET sites are different. If we average these results, the curve can be strange and hard to understand. Therefore, we decide to show the separate results.

- *Line 170: Can you provide an explanation for this dramatic decrease in SSA? In addition, for the Sahara area, SSA from the resolved method is much smaller than from the interpolated method. What explains this in terms of the aerosol size distribution? For example, is there a lot of fine-mode dust?*

  **Response:** We appreciate your insightful comments. We response this in your general comments above.

- *Line 171-173: The authors describe how AAOD is derived. Is this the same was as how AAOD is derived by AERONET?*

**Response:** Yes, both of them are $AAOD = AOD * (1 - SSA)$, the original description is kind of confusing for readers, so we add this in the revised version: "The AAOD is calculated by subtracting the scattering radiation from the extinction radiation (AOD) by aerosols, which can be described as AAOD=AOD*(1-SSA)"

- *Line 171: I think the reference should be to Figure 5 here.*

**Response:** Thank you for point our mistake out, it is corrected in the revised version.

- *Lines 206-207: The authors say that the TOA forcing is now more consistent with other studies for the Sahara. While Y Feng (2023) does show positive values over the Sahara (their Fig 11), the maximum values are close to 5 W/m-2 (which is also similar to Albani et al. 2014). Fig 7d has peak values that are 2-4 times larger. So there is a qualitative agreement, but not a quantitative one. Feng et al. (2022) has values > 10 W/m2. Feng et al. (2022) seem to describe the implementation of dust in the MPAS model, using the same methodology as in WRF-Chem. Section 2.2.6 in that paper does not describe the use of the "resolved" algorithm, so I assume the "interpolated" algorithm is used. But what is confusing is that MPAS produces a positive TOA forcing over the Sahara, but the WRF-chem simulation (Fig 7a) produces 0 to slightly negative values. Why? This statement seems to contradict what is stated in around line 206. In addition, other WRF-Chem studies show positive TOA forcing over the Sahara (e.g., DOI:10.1038/s41598-020-69223-4) which is presumably using the "interpolated" method. Not clear why different results are being obtained. It would be useful for authors to do more of a literature search on WRF-Chem used to predict radiative forcing over the Sahara.*

**Response:** We appreciate your thorough comment and the opportunity to clarify our findings. You are right that our agreement with Albani, et al. (2014) and Feng, et al. (2022) is qualitative rather than precisely quantitative. However, we believe this qualitative agreement is still meaningful for several reasons: The magnitude of dust radiative forcing is highly dependent on simulated dust mass loadings and spatial distributions. These factors can vary significantly between studies due to differences in model parameterizations, emission schemes, and meteorological conditions. Besides, the tuning factor for dust emissions is a significant source of uncertainty and can lead to substantial differences in results across studies. Given these inherent variabilities, we argue that qualitative agreement in the overall patterns and trends of dust forcing is a valuable validation of our methods.

Regarding the results in Feng et al. (2023), we would like to clarify that our two studies were conducted concurrently. The "Resolved" method was indeed used in Feng's work, although this was not explicitly stated in that paper. We are in the process of updating the descriptions in that paper to make this clear. We apologize for any confusion this may have caused. Feng et al., (2023) initially yielded a dust direct radiative forcing at the top of the atmosphere (TOA) of -0.75 W/m². This value significantly deviated from the observationally constrained estimate of -0.20 W/m² proposed by Kok et al. (2017). Subsequently, in the final version of Feng et al. (2023), we implemented the "Resolved" method, which resulted in a substantially improved estimate of -0.27 W/m². This marked improvement in alignment with observational constraints strongly suggests that the "Resolved" method demonstrates better performance in simulating radiative forcing, particularly for dust aerosols.

Concerning the results from DOI:10.1038/s41598-020-69223-4, it's important to note that their simulation period (May 22, 2006 to August 31, 2006) covers the boreal summer, when dust forcing at the top of the atmosphere (TOA) is typically stronger. This seasonal variation aligns with our own findings. In fact, our current study also

simulated positive TOA forcing in July 2015 for "Interpolated" method, as illustrated in the figure below:

[Figure]

Besides, the method for calculating dust radiative forcing is also different. In the study mentioned above, the dust radiative forcing is estimated using two sets of experiments, while in our study, it is diagnosed in a single experiment by running the radiation scheme several times. Our method is diagnosed aerosol direct radiative forcing, and in the other study, the results are estimated radiation perturbations due to dust that also includes the impacts of dust induced changes of clouds and circulation on radiation.

- *Lines 233-237: I understand it is useful to look at surface temperature. But 2-m observations are usually used to evaluate model predictions of temperature. I suspect that the impact on 2-m temperatures will be far less.*

**Response:** We appreciate your great suggestion, we analysis 2-m temperature instead of surface temperature in the revised manuscript. The impact on 2-m temperatures is a little less in some regions, but the discussion results remain the same.

- *Line 237: What are the other factors? Presumably altering aerosol-radiation interactions will also impact clouds which has direct effects and aerosol-cloud interactions could have other impacts as well. They mention these effects for Figure 10, but doesn't the same apply to Fig 9?*

**Response:** We appreciate that the reviewer points this out and sorry for the confusion. Actually, we want to express that the near-surface temperature is influenced not only by direct radiative effect and discuss other possible factors below. The discussion caused some confusion. We now make it clearer in the revised manuscript as: "This difference indicates that the radiative effects of aerosols on near-surface temperature are influenced not only by radiative forcing (direct radiative effect) but also by other factors such as aerosol-cloud interactions, the aerosols heating effects and the radiative effects on the surface radiative fluxes discussed below."

- *Line 255. This statement is hard to judge based on Figure S2. It would be better to show Figs S2c and S2f as "Resolved -ERA5" so that they can be directly compared to "Interpolated-ERA5". Also quoting some #'s on average bias over the domain (or subdomain) between the two simulations would be useful.*

**Response:** Thanks a lot for your suggestion regarding the presentation of Figure S2. We'd like to clarify our approach: We chose to present the difference between 'Resolved' and 'Interpolated' methods (Resolved - Interpolated) rather than comparing each method separately to ERA5 for a specific reason. The bias between our model results and ERA5 is substantially larger than the effects of the algorithm improvements on the simulation results. The possible reason could be the modelling biases in clouds in WRF-Chem, which further effect the near-surface temperature. We have a study on this phenomenon in progress. Presenting "Resolved - ERA5" and "Interpolated - ERA5" separately would make it difficult to discern the impact of our algorithmic changes due to this larger bias. By showing "Resolved – Interpolated", we can more clearly demonstrate that the "Resolved" method often has an opposite bias to "Interpolated - ERA5". This indicates that the "Resolved" method is actually reducing the overall bias compared to ERA5.

- *Line 266: I have the same comment for Figure S3 as Figure S2.*

**Response:** Thanks for your suggestion and please see the response for last comment.

- *Lines 298-300: Since the differences over China are small, it would be useful to speculate that they would also be small in other areas where anthropogenic aerosols dominate?*

**Response:** We appreciate your insightful suggestion to consider its implications for other regions. We add speculate this in revised manuscript as: "The discrepancies between the two algorithms show distinct regional characteristics. In China, where anthropogenic sources dominate the aerosol composition, the differences between the "Resolved" and "Interpolated" algorithms are relatively small, which could potentially be indicative of similar patterns in other regions dominated by anthropogenic aerosols."

**Reference**

Feng, J., Zhao, C., Du, Q., Xu, M., Gu, J., Hu, Z., and Chen, Y.: Simulating Atmospheric Dust With a Global Variable-Resolution Model: Model Description and Impacts of Mesh Refinement, Journal of Advances in Modeling Earth Systems, 15, e2023MS003636, https://doi.org/10.1029/2023MS003636, 2023.

Flanner, M. G., Liu, X., Zhou, C., Penner, J. E., and Jiao, C.: Enhanced solar energy absorption by internally-mixed black carbon in snow grains, Atmospheric Chemistry and Physics, 12, 4699–4721, https://doi.org/10.5194/acp-12-4699-2012, 2012.

Kok, J. F., Ridley, D. A., Zhou, Q., Miller, R. L., Zhao, C., Heald, C. L., Ward, D. S., Albani, S., and Haustein, K.: Smaller desert dust cooling effect estimated from analysis of dust size and abundance, Nature Geosci, 10, 274–278, https://doi.org/10.1038/ngeo2912, 2017.

Tan, J., Zhang, Y., Ma, W., Yu, Q., Wang, J., and Chen, L.: Impact of spatial resolution on air quality simulation: A case study in a highly industrialized area in Shanghai, China, Atmospheric Pollution Research, 6, 322–333, https://doi.org/10.5094/APR.2015.036, 2015.

Tao, H., Xing, J., Zhou, H., Pleim, J., Ran, L., Chang, X., Wang, S., Chen, F., Zheng, H., and Li, J.: Impacts of improved modeling resolution on the simulation of meteorology, air quality, and human exposure to PM2.5, O3 in Beijing, China, Journal of Cleaner Production, 243, 118574, https://doi.org/10.1016/j.jclepro.2019.118574, 2020.

Zhao, C., Ruby Leung, L., Easter, R., Hand, J., and Avise, J.: Characterization of speciated aerosol direct radiative forcing over California, Journal of Geophysical Research: Atmospheres, 118, 2372–2388, https://doi.org/10.1029/2012JD018364, 2013.

---

## Author Comment (AC2)

**Response to Referee #2**

*The study presents the updated version of the aerosol radiative feedback algorithm in the coupled meteorology-chemistry model WRF-Chem v4.4. The manuscript also describes the current algorithm utilized in the default version of WRF-Chem. A full gas-aerosol chemistry scheme CBMZ-MOSAIC is used to simulate the aerosol radiation feedback for different regions of the world. One domain covers China, and another one covers the Saharan region.*

*The model simulations with the default and updated algorithms show some differences in the aerosol optical depth and single scattering albedo, in particular for the Saharan region. The authors show better agreement of the model with the AERONET AOD data when the updated aerosol radiative feedback algorithm is used.*

*The updated algorithm can help to improve the WRF-Chem air quality and meteorological simulations. I recommend publishing the manuscript after addressing the following comments.*

**Response:** We sincerely appreciate your thorough and insightful review of our manuscript. Your comments and suggestions have been invaluable in enhancing the quality and clarity of our work. We have addressed each of your specific comments, as detailed in our point-by-point responses below. We believe these revisions have strengthened our manuscript and hope that they adequately address your concerns.

- *Lines 235-240: You mention the indirect feedback. Are you referring to semi-direct feedback? I assume your WRF-Chem configuration does not include the aerosol-cloud interactions.*

**Response:** We appreciate that you point this out and sorry for the confusion. Yes, we refer to semi-direct feedback and we modified our description as: "these effects include not only the direct radiative effect (DRE) of aerosols but also the indirect consequences: the alterations in radiative forcing induced by aerosols can lead to effects on the atmospheric energy balance, which in turn influence other meteorological processes such as the cloud formation and thus further affecting radiation." Besides, our WRF-Chem configuration includes the aerosol-cloud interactions. The aerosols spatial distributions and physical properties can interact with the Morrison 2-momnet microphysics scheme, which we used in WRF-Chem.

Actually, we want to express that the near-surface temperature is influenced not only by direct radiative effect and discuss other possible factors below. The discussion caused some confusion. We now make it clearer in the revised manuscript as: "This difference indicates that the radiative effects of aerosols on near-surface temperature are influenced not only by radiative forcing (direct radiative effect) but also by other factors such as aerosol-cloud interactions, the aerosols heating effects and the radiative effects on the surface radiative fluxes discussed below."

- *The resolution of your WRF-Chem grid is quite coarse (50km). I'm wondering if high-resolution simulations (e.g. 4-10km) would yield somewhat different results. For instance, anthropogenic pollution can be better simulated by a high-resolution model.*

**Response:** We appreciate your insightful comment regarding our model resolution and the potential implications of using higher-resolution simulations. Our choice of a 50km grid resolution was based on several considerations: Computational resources: Given the spatial domain and extended time period of our simulations, a 50km resolution allowed us to complete our study within available computational resources; Besides, our primary aim was to investigate the effects of modifications of algorithm on simulating results. Simulating an accurate aerosols distribution is not our main purpose. Therefore, we consider that a 50km resolution is generally adequate for this study.

We acknowledge that higher-resolution simulations (e.g., 4-10km) could indeed yield somewhat different results, particularly for anthropogenic pollution. Potential benefits of higher resolution include: Improved representation of local emissions: Finer grids can better resolve point sources and urban areas; Enhanced simulation of meteorological processes: Higher resolution can better capture local circulations and boundary layer processes that affect pollutant transport and transformation; More accurate terrain representation: This could improve simulations in areas with complex topography and further effect the physical processes of aerosols such as emission, deposition and transport processes.

Given these potential impacts from resolution, we agree that it would be valuable to explore how higher-resolution simulations will improve the simulation results of aerosols distributions. However, our main purpose is to demonstrate the importance of using a spectrally resolved method for calculating aerosol optical properties, rather than to improve the overall model performance of aerosol distributions. We have added a paragraph in the discussion section of our manuscript acknowledging the model resolution's effects: "It's important to note that the primary aim of our study is to demonstrate the importance of using a spectrally resolved method for calculating aerosol optical properties, rather than to improve the overall model performance. Many factors can affect the simulation of meteorological fields and radiative processes beyond the optical properties methods we're investigating in this study. For example, while our study employs a 50km grid resolution, which is suitable for investigating aerosol-radiation interactions, higher-resolution could enhance the simulation of aerosol emission, deposition, and transport processes, potentially leading to a more accurate representation of aerosol distributions and their radiative effects (Feng et, al., 2023; Tan et, al., 2015; Tao et, al., 2020). In addition to model resolution, the calibration factors for aerosol emission rates, the representation of aerosol size distributions, the quality and accuracy of input data, and the selection and implementation of parameterization schemes for various physical processes could all introduce uncertainties and potential impacts on the simulated results."

- *I recommend adding the satellite AOD data for the evaluation of the model sensitivity simulations. The spatial coverage of the satellite observations would*

*greatly complement the sparse AERONET observations used in the model evaluations.*

**Response:** We thank the reviewer for inquiring about the use of other datasets. MODIS satellite AOD is typically available only at 550 nm. For the 'Interpolated' and 'Resolved' methods, the AOD results are similar at 550 nm because both methods use the same calculation approach when the wavelength is the same. While satellite results could cover more regions, the comparison at a single wavelength would not effectively demonstrate the differences between our methods. The multi-wavelength data from AERONET allows us to show these differences more clearly. In the revised manuscript, we add these discussions: "To evaluate the modelling results, multi-spectral aerosol optical depth (AOD) measurements are required, which is critical for this study comparing the "Interpolated" and "Resolved" methods across wavelengths. Therefore, the retrieved total AOD from the AERONET network (Holben et al., 1998). Although satellite AOD products such as from the Moderate Resolution Imaging Spectroradiometer (MODIS) have greater spatial coverage, the number of wavelengths is limited. Comparison at only one or very few wavelengths would not effectively demonstrate the distinctions between the two algorithms examined in this work."

- *Given the significant impact of the updated algorithm on heating rates over Sahara, it'd be helpful to evaluate the temperature simulations by WRF-Chem. Such an evaluation would help to determine whether the improved AOD and SSA simulations lead to improvement of the meteorological simulations. The authors present some comparisons with the reanalysis data, but comparing directly to the surface stations (e.g. 2m air temperature) would be really informative. To my knowledge, some global models such as GFS don't assimilate the ground-based weather observations.*

**Response:** Thanks for your great suggestion. While we understand the value of such comparisons, we would like to address this concern and explain our approach:

Focus of the study: The primary aim of our study is to demonstrate the importance of using a spectrally resolved method for calculating aerosol optical properties, rather than to improve the overall model performance. We show that the "Resolved" method can capture complex relationships between aerosols' optical properties and wavelengths that the "Interpolated" method may miss, particularly for dust-dominated regions and at specific wavelengths. Besides, our results demonstrate that the amendment of algorithms can significantly affect the simulation results. While these changes may not necessarily lead to better agreement with observations in all cases, they give the model more potential to improve simulation abilities by more accurately representing the underlying physical processes.

Complexity of model evaluation: It's important to note that many factors can affect the simulation of meteorological fields and aerosol properties beyond the optical properties methods we're investigating such as model resolution, physical algorithms, parameterization schemes, and input data quality. Given this complexity, direct comparisons of the "Resolved" and "Interpolated" methods with observations may not

provide a conclusive assessment of whether our modifications improve the model's overall simulation abilities.

We have added further discussions in Section 4 as: "It's important to note that the primary aim of our study is to demonstrate the importance of using a spectrally resolved method for calculating aerosol optical properties, rather than to improve the overall model performance. Many factors can affect the simulation of meteorological fields and radiative processes beyond the optical properties methods we're investigating in this study. For example, while our study employs a 50km grid resolution, which is suitable for investigating aerosol-radiation interactions, higher-resolution could enhance the simulation of aerosol emission, deposition, and transport processes, potentially leading to a more accurate representation of aerosol distributions and their radiative effects (Feng et, al., 2023; Tan et, al., 2015; Tao et, al., 2020). In addition to model resolution, the calibration factors for aerosol emission rates, the representation of aerosol size distributions, the quality and accuracy of input data, and the selection and implementation of parameterization schemes for various physical processes could all introduce uncertainties and potential impacts on the simulated results. Given this complexity, direct comparisons of the "Resolved" and "Interpolated" methods with observations may not provide a conclusive assessment of whether our modifications improve the model's overall simulation abilities. Therefore, we didn't evaluate the model's simulation results of meteorological fields from two methods by comparing with more observation results other than the ERA5 reanalysis dataset. Our results show that the "Resolved" method can capture complex relationships between aerosols' optical properties and wavelengths that the "Interpolated" method may miss, particularly for dust-dominated regions and at specific wavelengths where water contents have a significant larger absorption than other wavelengths. Besides, our results demonstrate that the amendment of algorithms can significantly affect the simulation results of meteorological fields. While these changes may not necessarily lead to better agreement with observations in all cases, they give the model more potential to improve simulation abilities by more accurately representing the underlying physical processes."

**Reference**

Feng, J., Zhao, C., Du, Q., Xu, M., Gu, J., Hu, Z., and Chen, Y.: Simulating Atmospheric Dust With a Global Variable-Resolution Model: Model Description and Impacts of Mesh Refinement, Journal of Advances in Modeling Earth Systems, 15, e2023MS003636, https://doi.org/10.1029/2023MS003636, 2023.

Tan, J., Zhang, Y., Ma, W., Yu, Q., Wang, J., and Chen, L.: Impact of spatial resolution on air quality simulation: A case study in a highly industrialized area in Shanghai, China, Atmospheric Pollution Research, 6, 322–333, https://doi.org/10.5094/APR.2015.036, 2015.

Tao, H., Xing, J., Zhou, H., Pleim, J., Ran, L., Chang, X., Wang, S., Chen, F., Zheng, H., and Li, J.: Impacts of improved modeling resolution on the simulation of meteorology, air quality, and human exposure to PM2.5, O3 in Beijing, China, Journal of Cleaner Production, 243, 118574, https://doi.org/10.1016/j.jclepro.2019.118574, 2020.